# Dynamic label-free analysis of SARS-CoV-2 infection reveals virus-induced subcellular remodeling

Nell Saunders[1], Blandine Monel[1], Nadège Cayet[2], Lorenzo Archetti [3,6], Hugo Moreno [3,6], Alexandre Jeanne [3,6], Agathe Marguier [3], Julian Buchrieser [1], Timothy Wai [4], Olivier Schwartz [1,5] ✉ & Mathieu Fréchin [3] ✉

Assessing the impact of SARS-CoV-2 on organelle dynamics allows a better understanding of the mechanisms of viral replication. We combine label-free holotomographic microscopy with Artificial Intelligence to visualize and quantify the subcellular changes triggered by SARS-CoV-2 infection. We study the dynamics of shape, position and dry mass of nucleoli, nuclei, lipid droplets and mitochondria within hundreds of single cells from early infection to syncytia formation and death. SARS-CoV-2 infection enlarges nucleoli, perturbs lipid droplets, changes mitochondrial shape and dry mass, and separates lipid droplets from mitochondria. We then used Bayesian network modeling on organelle dry mass states to define organelle cross-regulation networks and report modifications of organelle cross-regulation that are triggered by infection and syncytia formation. Our work highlights the subcellular remodeling induced by SARS-CoV-2 infection and provides an Artificial Intelligence-enhanced, label-free methodology to study in real-time the dynamics of cell populations and their content.

The COVID-19 pandemic is caused by the severe acute respiratory syndrome-coronavirus-2 (SARS-CoV-2)[1], inducing a broad spectrum of syndromes from a light cold to life-threatening pneumonia[2]. The search for SARS-CoV-2 treatments is continuing[3] and reductionist approaches vastly dominate experimental efforts. A stop-motion view of the SARS-CoV-2 infection cycle has emerged[4], where its impact on the host cell is understood through key host/virus molecular entanglements[5]. Previous studies tackled the impact of the virus on a global cellular scale, employing fluorescence and electron microscopy[6,7], and as, such were lacking the dimension of time. Filming the impact of SARS-CoV-2 on an entire cellular system from early infection to death would greatly improve our understanding of infection sequences and dynamics, yet the efforts to obtain such knowledge are precluded by the limitations of live microscopy. The various types of fluorescence microscopy induce non-neglectable phototoxicity and molecular perturbations, due to the use of chemical or genetic labeling[8–13]. This limits the capacity to observe multiple targets over hours-long periods, which is the time scale necessary to capture the cellular changes induced by SARS-CoV-2. Classical label-free imaging techniques such as phase contrast or differential interference contrast (DIC), while less invasive, provide images plagued by optical aberrations, poor contrast, and limited spatial resolution. AI-augmented label-free microscopic methods have emerged[14], emulating fluorescence staining for key cellular structures in the absence of fluorescence[15,16] or bolstering the usage of lower-content, label-free images to detect specific cellular states[17–19]. Holotomographic

[1]Virus & Immunity Unit, Institut Pasteur, Université Paris Cité, CNRS, UMR 3569 Paris, France. [2]Institut Pasteur, Université Paris Cité, Ultrastructural Bioimaging Unit, 75015 Paris, France. [3]Deep Quantitative Biology Department, Nanolive SA, Tolochenaz, Switzerland. [4]Mitochondrial Biology Group, Institut Pasteur, Université Paris Cité, CNRS, UMR 3691 Paris, France. [5]Vaccine Research Institute, Creteil, France. [6]These authors contributed equally: Lorenzo Archetti, Hugo Moreno, Alexandre Jeanne. ✉e-mail: Olivier.schwartz@pasteur.fr; Mathieu.frechin@nanolive.ch

microscopy (HTM) provides high-content refractive index (RI) images able to capture complex biological processes and multiple cellular structures at unprecedented spatial resolution and ultralow-power illumination[20]. When combined with computer vision, HTM can support image-based quantitative investigations of cell dynamics over hours at relevant temporal resolutions[20].

In this study, we developed a high-content imaging pipeline combining live HTM, machine learning, and Bayesian network modeling to provide a quantitative and dynamic vision of the impact of SARS-CoV-2 on the organelle system of hundreds of infected cells in culture.

## Results

### Label-free microscopy shows virus-induced cellular alterations

Through key host–viral protein interactions, SARS-CoV-2 reshapes the subcellular organization and the organelles of its target cells[6,7]. SARS-CoV-2 reroutes lipid metabolism[21–23], fragments the Golgi apparatus[7], promotes the formation of double-membrane vesicles[7,24,25] and alters mitochondrial function[26], with the goal of boosting virus production while delaying antiviral responses[27,28]. Our aim was to capture the kinetics and extent of such alterations in living cells by recording and quantifying cellular and organellar dynamics in real time using HTM[20]. We selected U2OS-ACE2 cells as targets because of their high sensitivity to SARS-CoV-2 and their flat shape that facilitates imaging[29]. Cells were first infected with the Wuhan strain and imaged with HTM (Fig. 1A and Supplementary Fig. 1a, b). Time-lapse experiments were carried out for up to 2 days or until the death of infected cells (Fig. 1B). Non-infected cells were recorded as a control. The most obvious event visible as soon as 10 h post-infection (pi) was the formation of syncytia, a known phenomenon where infected cells expressing the viral spike (S) protein at their surface fuse with neighboring cells[29,30].

Formation of syncytia was used as a marker of productively infected cells. In such cells, we noticed a quick clustering of nuclei, visible as soon as two or more cells started to fuse. The zone of nuclei clustering apparently hosted groups of growing lipid droplets (LD), accumulating over time, while mitochondria were moving away from this region and redistributed across the cytoplasm. Within the nuclei, nucleoli appeared denser and rounder upon infection (Fig. 1C). We next determined which of these cellular events were due to the infection itself or the result of syncytia formation. To differentiate between these possibilities, we recorded cells that fused together in the absence of SARS-CoV-2, after transient expression of Syncytin-1, a fusogenic protein involved in the formation of placental syncytiotrophoblasts[31]. Infection-independent syncytia did not show the same features. LD remained small and rare, and nucleoli were not altered (Fig. 1B and C).

However, mitochondrial movements in both SARS-CoV-2- and Syncytin-1-induced syncytia seemed globally similar. As demonstrated before[20], we did not detect the Golgi network, the cytoskeleton, or DMVs with HTM since these structures show little RI contrast with their surroundings.

To go beyond the qualitative nature of these observations and to quantify the cellular alterations triggered by SARS-CoV-2, we designed an HTM image quantification pipeline in which cells and organelles were detected in time-lapse recordings by tailored machine learning (ML) approaches (Fig. 1D).

### Machine learning detects cellular organelles in high-resolution label-free images

We adapted our ML strategy to the different characteristics of the biological objects of interest. Mitochondria that are small, pixel-scale objects were detected using a two-class pixel categorization[32], where a trained extra-tree classifier attributes a class to each pixel based on its position in a derived feature space. Such an approach allowed precise and accurate detection of these sparse objects within the highly textured HTM images (Fig. 2A). Its large hyperparameter space was not explored through a *human-in-the-loop* process but using the optuna optimization framework[33]. Larger objects such as nuclei, nucleoli, and whole cells were optimally segmented with an adapted U-NET[34] fully convolutional network (Fig. 2B, Supplementary Fig. 2). For whole cell segmentation, a sharpening of the outlines was performed by object propagation within the RI signal[35]. LD was segmented using Nanolive's dedicated assay, which automatically detects LD based on their high refractive index, unique signal distribution, and roundness.

We then validated the automatic segmentations of organelles within RI images by labeling the cells with organelle-specific fluorescent markers. Nuclei, nucleoli, LD, and mitochondria were stained respectively with Hoechst, Green Nucleolar staining (ab139475), lipid spot, and Mitotracker DeepRed (Fig. 2C–G, Supplementary Fig. 3a–d). We used the standard F1 and intersection over union (IoU) scores for the strict binary evaluation of masks versus references and the structural similarity index measure (SSIM)[36] for a quantification of similarity perception. For nuclei and nucleoli that are large and simple objects, matches between masks and fluorescent signals were high (Figs. 2C, G and 3B, Supplementary Fig. 3a). The LD masks did not perfectly match with the lipid spot fluorescent signal. This was expected since HTM resolving power is better than epifluorescence[20] (Fig. 2E, Supplementary Fig. 3c). This illustrates the challenge of objectively quantifying the quality of few-pixel object masks, especially in a live context where biological structures move through the succession of acquisition regimes. For these reasons, the scores of LD predictions were very good yet lower than those of nuclei and nucleoli (Fig. 2G).

Similarly, our RI-based mitochondrial predictions were sharper and better resolved than the fluorescent signal generated by Mitotracker DeepRed (Fig. 2F, Supplementary Fig. 3d). The scores obtained from comparing our RI-based ML predictions against fluorescence-derived references were good (Fig. 2G) yet lower than those of the other organelles because of unavoidable mismatches between predictions and ground truth. In addition to organelle motion, the typical crowding of mitochondria in the perinuclear region generates unresolved[37] Mitotracker epifluorescence signal. Such a signal is not optimal for comparison purposes. We thus used an expert-generated segmentation of mitochondria within an RI image to assess further the quality of our ML-generated mitochondrial mask (Fig. 2G).

To validate the biological relevance of our mitochondrial detection workflow, we silenced *OPA1*, a dynamin-like GTPase protein required for mitochondrial fusion[38], whose ablation causes mitochondrial fragmentation and inherited optic neuropathy[39]. Inspection of the label-free HTM images revealed an obvious fragmentation of the mitochondrial network and alteration of mitochondria shapes, and our ML-based mitochondrial detection system reported a reduction of mitochondria size distribution (Supplementary Figs. 4 and 5). This confirmed our capacity to automatically detect and quantify a broad range of mitochondrial morphology under basal and pathological conditions in a label-free manner.

### SARS-CoV-2 infected cells display specific organelle dynamics

We then examined in real-time the subcellular changes induced by infection with two different SARS-CoV-2 strains. We recorded movies of U2OS cells infected with either the SARS-CoV-2 Wuhan ancestral virus or the Omicron BA.1 variant. As controls, we used uninfected cells and cells that underwent intercellular fusion upon Syncytin-1 expression. We used our algorithms to identify over time single or fused cells with a precise segmentation even in time of transition from the single cell to syncytium state (Supplementary Fig. 6 and Supplementary Movies 1–19) and to segment nuclei, nucleoli, LD, mitochondria, and the cytosol. The cytosol is defined by the removal of all organelle segmentations from the single cells or syncytium segmentations. (Supplementary Fig. 6 and Supplementary Movies 1–19). This analysis was performed every 12 min, from 2 to 48 h pi.

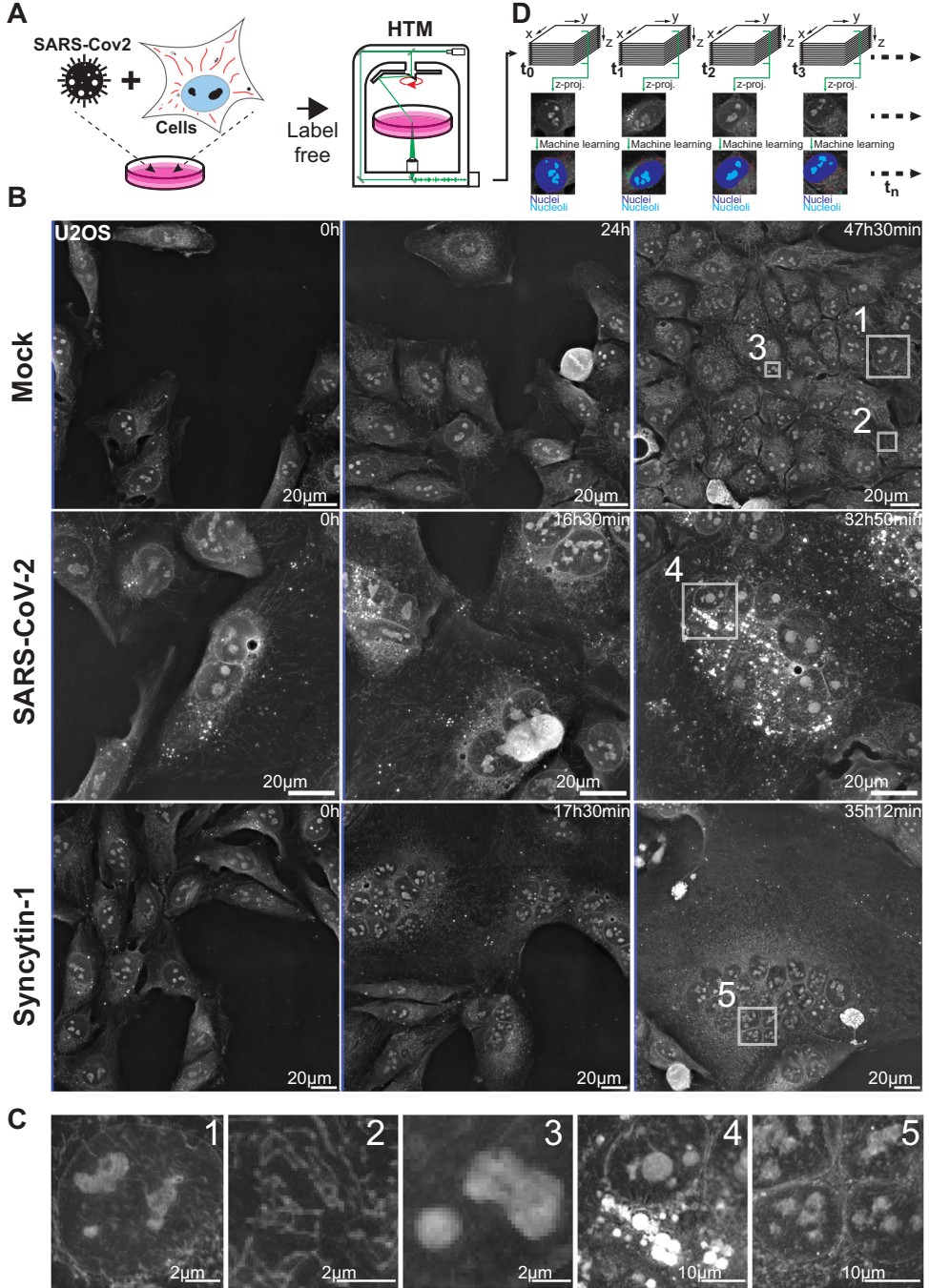

**Fig. 1 | SARS-Cov-2 infected cells show subcellular dynamic changes.**
**A** Refractive index (RI) map of U2OS cells infected with SARS-Cov-2, was acquired using holotomographic microscopy (HTM). **B** Representative images of non-infected cells (Mock), SARS-CoV-2 infected cells, or cells forming syncytia upon expression of the fusogenic protein Syncytin-1 (# of independent time-lapse acquisitions ≥ 4). **C** Magnifications of cellular details available for further ML-aided image analysis, such as nuclei (1), mitochondria (2), nucleoli (3), lipid droplets (4), or syncytia nuclei cluster (5). **D** Outline of our time-lapse imaging analysis pipeline ensuring data processing, computer vision, and quantitative assessment of cells.

Our object segmentation was robust, allowing analysis over time within each movie, and across movies (Fig. 3 and Supplementary Movies 1–19). Of note, the formation of syncytia did not alter the quality of our predictions. Nuclei and nucleoli were properly detected despite the appearance of compact clusters of nuclei in syncytia (Fig. 3A and B). LD remained well-defined even when their size increased or when they moved from the cytosol to the perinuclear region (Fig. 3C and D and Supplementary Fig. 7). Mitochondria were accurately segmented, despite a large variety in length and distribution in infected cells (Fig. 3C and D).

We represented the data as time series of violin plots based on single cells or organelles points, to visualize the evolution of the various parameters over time and to allow explicit statistical assessment of infection-induced changes (Fig. 4). The progression of nucleoli/nucleus size ratio was similar in Wuhan-infected and non-infected cells, as well as in syncytia triggered by Syncytin-1 (Fig. 4A). In agreement with a recent report[40], the nucleoli of Omicron-infected cells became larger, especially at a late time point of infection. This might reflect the recruitment of the nucleolar machinery to facilitate viral translation[41,42]. A massive

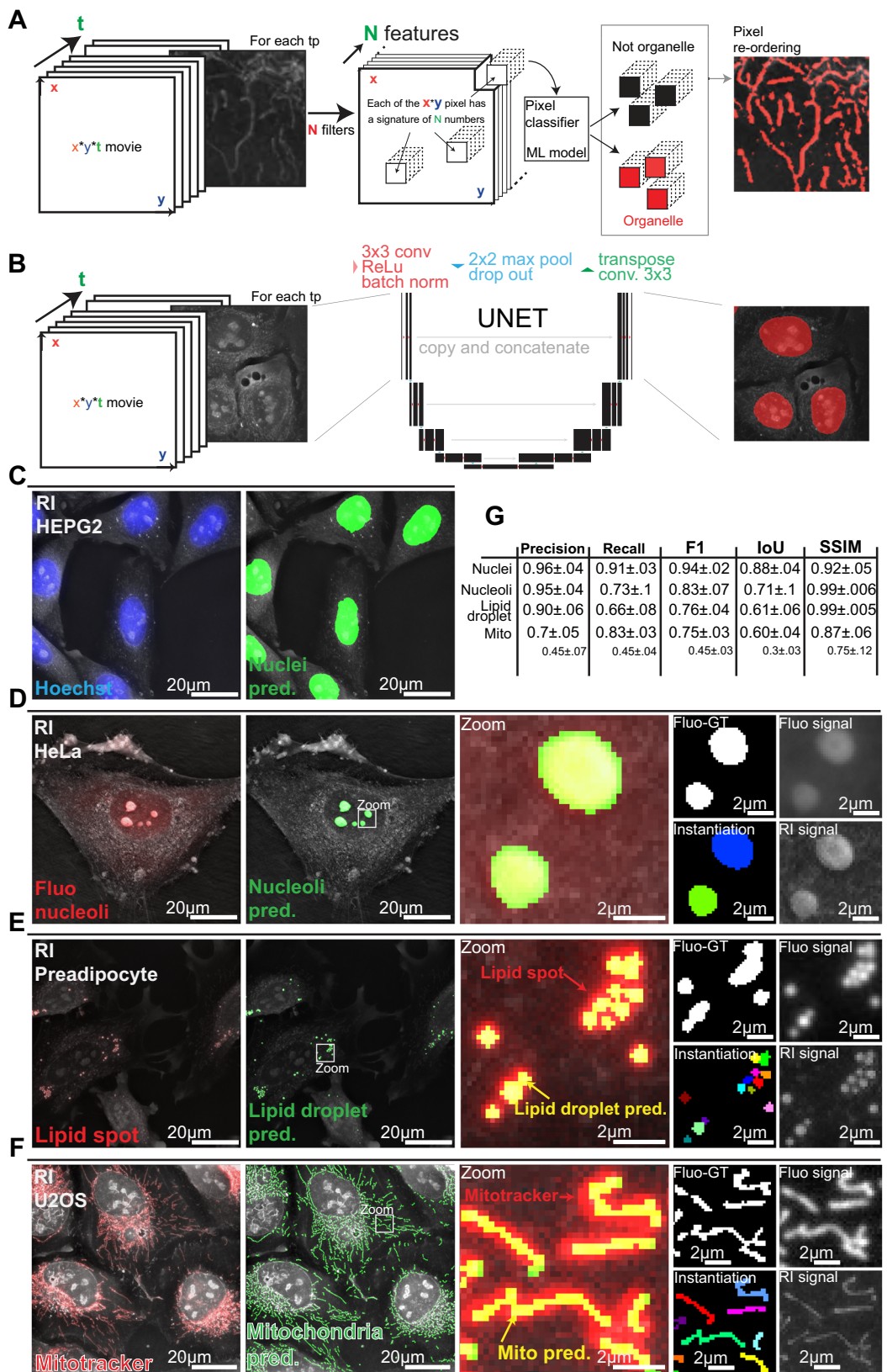

| | Precision | Recall | F1 | IoU | SSIM |
|---|---|---|---|---|---|
| Nuclei | 0.96±.04 | 0.91±.03 | 0.94±.02 | 0.88±.04 | 0.92±.05 |
| Nucleoli | 0.95±.04 | 0.73±.1 | 0.83±.07 | 0.71±.1 | 0.99±.006 |
| Lipid droplet | 0.90±.06 | 0.66±.08 | 0.76±.04 | 0.61±.06 | 0.99±.005 |
| Mito | 0.7±.05 | 0.83±.03 | 0.75±.03 | 0.60±.04 | 0.87±.06 |
| | 0.45±.07 | 0.45±.04 | 0.45±.03 | 0.3±.03 | 0.75±.12 |

increase in LD number and size was observed in SARS-CoV-2 Wuhan- or Omicron-infected cells but not in control or Syncytin-1-expressing cells (Fig. 4B). These observations are in line with SARS-CoV-2-induced remodeling of lipid metabolism[21] to support pro-virus signaling[23] and provide material for the formation of double-membrane vesicles (DMV) (Fig. 4B).

The obvious effect of SARS-CoV-2 on LD indicated a broad metabolic impact and thus led to the question of mitochondrial alterations. In control cells, we detected a modest yet significant reduction of mitochondrial size over the 1.5 days of culture, which may reflect the changing metabolic state of proliferating cells that progressively consumed culture medium (Fig. 4C). In both Wuhan and

**Fig. 2 | Machine learning detects key organelles within HTM images.** RI-refractive index, Mito pred-mitochondria predictions. **A** Mitochondria and lipid droplets detection using pixel classification: a feature space of size $x*y*N$ is calculated by applying $N$ convolution filters on each image time point (tp) of size $x*y$. An extra tree classifier decides for each pixel if it belongs or not to an organelle signal based on its position in the feature space. **B** Nuclei, nucleoli, and cells detection using the convolutional network UNET. **C–F** Comparison of **C** nuclei, **D** nucleoli, **E** lipid droplet, and **F** mitochondria detection within refractive index images with their respective fluorescent label signal made partially transparent such that underlying cellular structures are visible. Structures are thicker when visualized with epifluorescence microscopy compared to holotomography. **G** Precision, recall, and F1 score, as well as intersection over union (IoU) score and structural similarity index measure (SSIM) for each organelle. Nuclei and nucleoli prediction scores are calculated against fluo-derived ground truths (Fluo-GT). Lipid droplets and mitochondria are evaluated against expert-corrected or raw fluo-derived ground truth.

Omicron infected cells, there was a marked mitochondria size decrease around 15 h (900 min) pi (Fig. 4C), that is likely an infection-induced imbalance of mitochondrial dynamics[43]. At later points, the length of the mitochondria increased again in infected cells. Contradictory findings of impaired mitochondrial fission and fusion in response to SARS-CoV-2 infection have been reported[44–46]. Our results suggest that these events may be temporally distinct.

We then quantified the dry mass, defined as the bulk content in biomolecules (mainly proteins, lipids, and nucleic acid) that are not the water of hundreds of thousands of mitochondria. In fact, HTM returns the refractive index of the observed biological structures, which is linearly linked to the content in biomolecules of the observed structure. In control cells, we observed a stable dry mass per unit of mitochondria size overtime (Fig. 4D). This dynamic was changed in SARS-CoV-2-infected or Syncytin-1-expressing syncytia, which were characterized by an overall slight reduction of the dry mass of single mitochondria. This is in line with previous reports[46,47] suggesting that SARS-CoV-2 down-regulates the translation of mitochondrial genes. The observation that Syncytin-1-expressing cells also displayed a decreased mitochondrial dry mass suggests that SARS-CoV-2 could use syncytia formation to promote mitochondrial alterations more efficiently to promote mitochondrial alterations more efficiently through molecular and structural mechanisms that still need to be discovered.

We next took advantage of our capacity to localize organelles in space and relative to each other to measure the distance between LD and mitochondria, a marker of the rate of fatty acid oxidation and thus of energy production[33,34]. In uninfected or Syncytin-1-expressing cells, the proportion of LD in proximity (<400 nm) of mitochondria increased over time (Fig. 4E). In contrast, this ratio stayed stable or even decreased in Wuhan and Omicron-infected cells. Therefore, infection separates LDs from mitochondria, reflecting a probable impact of SARS-CoV-2 on cell metabolism.

**Large lipid droplets form in infected cells only**

We next examined the links that may exist between LD, mitochondria, and viral production zones, which we identified by immuno-staining with anti-NSP3 or anti-double stranded (ds) RNA antibodies. The viral NSP3 protein plays many roles in the virus life cycle, including viral polyprotein processing, formation of the viral replication compartment, viral RNA trafficking, and innate immunity antagonism[48]. Anti-dsRNA antibody specifically recognizes double-stranded RNA (dsRNA) of >40 bp in length generated during the replication of positive sense genome viruses and thus selectively recognizes viral RNAs and not cellular RNAs[49], validated further by the absence of dsRNA signal in cells that do not display an NSP3 signal (Fig. 5A–C and Supplementary Fig. 7). NSP-3 and dsRNA, always both present in single cells showing fluorescent signal (Fig. 5A–D), rarely colocalized at the subcellular level as demonstrated by colocalisation experiments and by the absence of correlation of both signal shown by a Pearson coefficient of 0.34 (Fig. 5B and C). Correlative HTM and confocal microscopy show that at 24 h pi, cells that accumulated large perinuclear LD were also positive for dsRNA and NSP3 (Fig. 5A and Supplementary Fig. 3), confirming that the accumulation of large LD is a signature of SARS-CoV-2 infection. dsRNA accumulated next to LD, while NSP3 was excluded from them (Fig. 5B–D). We performed correlative HTM and electron microscopy (CHEM) of non-infected or infected cells and focused on areas displaying large LD. The mitochondria surrounded large LD in control but not in SARS-CoV-2 infected cells (Fig. 5E, F and Supplementary Fig. 4). Altogether, the HTM quantitative analysis, combined with qualitative correlative microscopy indicates that LD alteration together with mitochondria relocation is a hallmark of effective virus production.

**SARS-CoV-2 alters organelles cross-regulations**

We next investigated the dependencies between LD, mitochondria, nuclei, nucleoli, and cytosol dry masses using Bayesian networks (BN) established on ten thousand different datasets of the same size as the original dataset, created through the random sampling of the original dataset while allowing replacement. BN is an established tool for modeling biological datasets[50] in fields such assignaling[51], genomics[52,53], or immunology[54], and to model protein-nuclear bodies interactions[55], but not, to the best of our knowledge, to model the hierarchy and regulation existing between cell organelles. Established BN methods[56] allows us to search for the direct dependencies (while filtering out spurious correlations) and probabilities between factors of interest. These methods provide an intuitive visual representation in the form of a directed acyclic graph. We thus investigated, over all the bootstrapped datasets, the probability of discretized organelles' dry mass state, given the co-existing dry mass of the other organelles. We then reported under the form of inter-organelle networks the occurrence of specific directional dependencies. Considering that the dry mass variation of a subcellular compartment reflects regulated variations of its protein, lipid, and/or nucleic acid contents, we propose that the relationship between organelle dry mass captures an integrated level of organelle-dependent regulation. Henceforth, we will employ the term *organelle cross-regulation* (OCR) rather than referring to *dry mass influence diagrams*, the latter being less intuitive. As our approach relies on organelle detection and not on gene or protein level measurements, it is blind to organelle-independent global regulations that manifest during infection or cellular dysfunction, such as modulations of protein expression and other gene regulations that cannot be captured by assessing dry mass variations.

We also attributed a "regular" or "syncytium" identifying tag, to each cell of our control, SARS-COV-2-infected and Syncytin-1 expressing conditions. This allowed us to observe the specific impact of syncytia formation on OCR compared to infection. The differences between similarly established networks for the three conditions provide unique insights into how SARS-CoV-2 infection or syncytia formation impacts OCR (Fig. 6A–C). The nucleus has a strong influence on nucleoli, in all conditions, irrespective of infection or syncytia formation. This indicates that our networks can capture relevant functional relationships between organelles. Moreover, the nucleus dry mass sits in the center of the unperturbed OCR, which is expected since, in freely dividing cells, the cell cycle and thus the nucleus DNA content, synchronizes the rest of the OCRs (Fig. 6A).

We observed that SARS-CoV-2 infection rewired the OCR network (Fig. 6B), which can be explained by the fact that it can alter organelles involved in numerous processes, such as lipid metabolism[21–23] and the cell cycle[57–60]. When compared to the unperturbed condition (Fig. 6A), one can observe that the infection-induced OCR (Fig. 6B) lost the link between the nucleus and LDs, and mitochondria and LDs, confirming

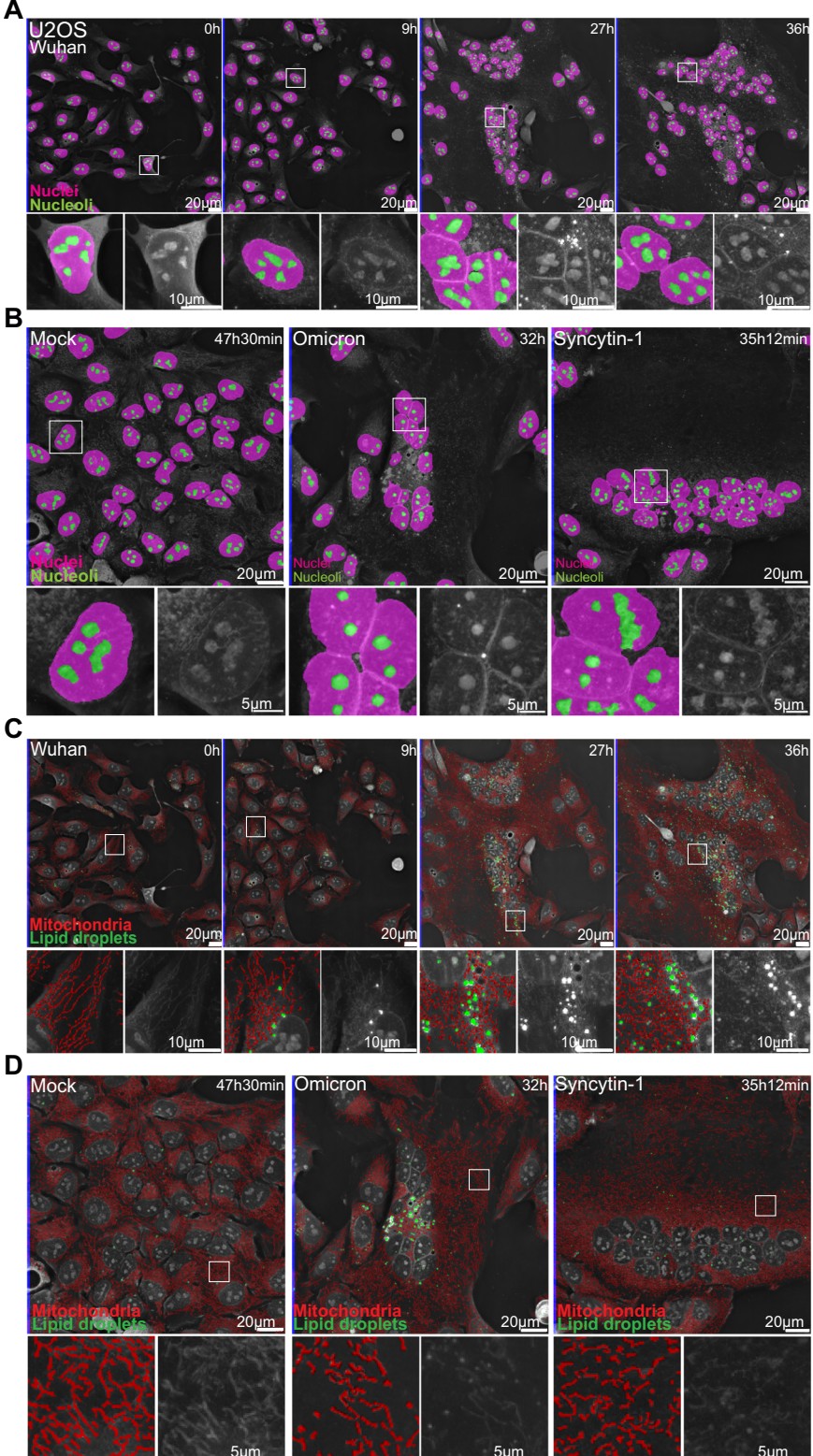

**Fig. 3 | Automated detections of cellular organelles capture dynamics of SARS-CoV-2-induced changes. A** 36 hours-long holotomographic microscopy (HTM) time-lapse acquisition of U2OS cells infected by the Wuhan SARS-CoV-2 strain. Pink: nuclei. Green: nucleoli. **B** Late time point images of time-lapse imaging experiments of non-infected cells (Mock), Omicron-infected cells, and Syncytin-1-expressing cells. Pink: nuclei. Green: nucleoli. **C** 36 hours-long holotomographic microscopy

(HTM) time-lapse acquisition of U2OS cells infected by the Wuhan SARS-CoV-2 strain. Red: mitochondria. Green: lipid droplet. **D** Late time point images of time-lapse imaging experiments of non-infected (Mock), Omicron-infected, and Syncytin-1-expressing cells. Red: mitochondria, Green: LD (# of independent time-lapse acquisitions ≥ 4).

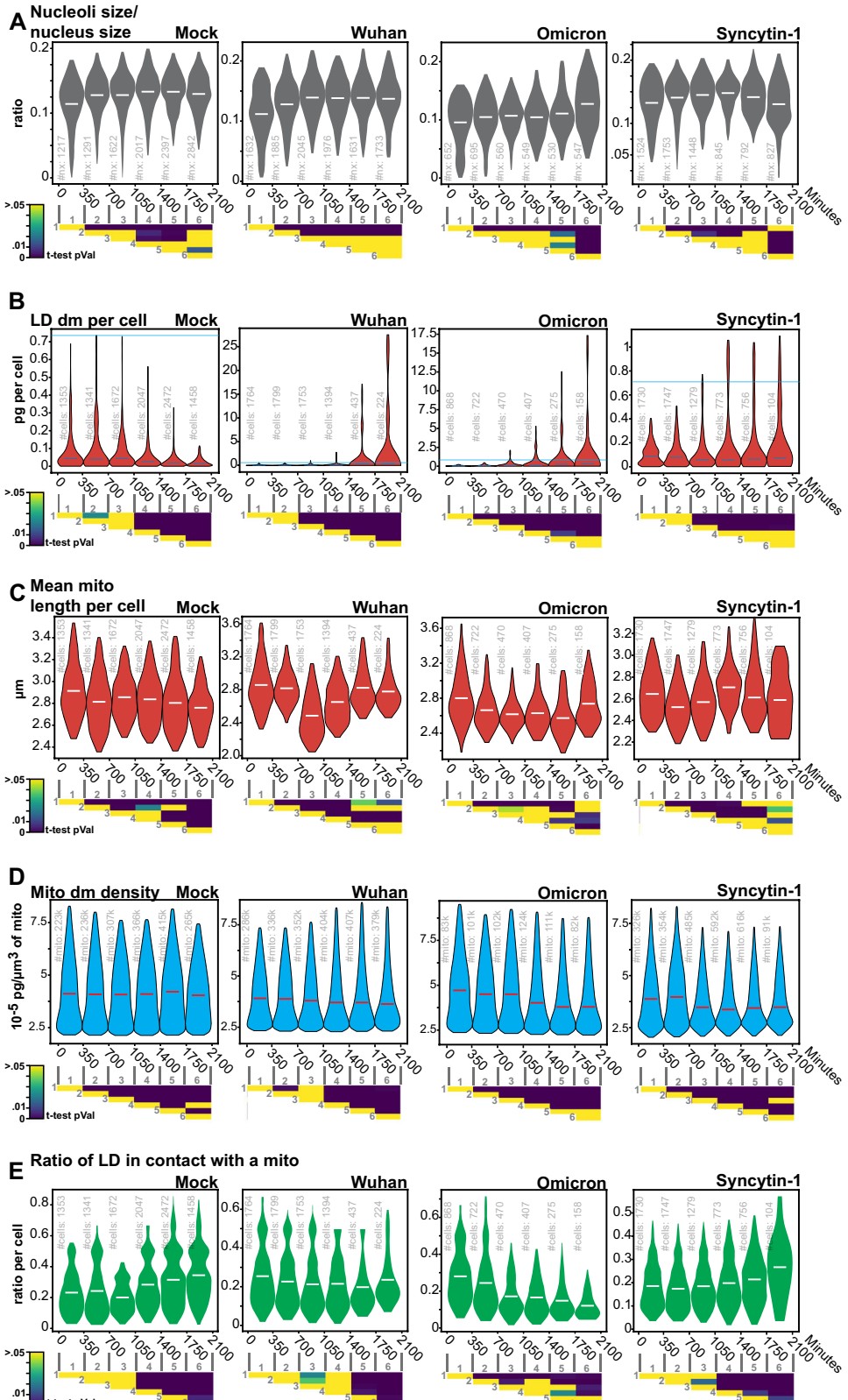

**Fig. 4 | Label-free quantifications of SARS-CoV-2-induced alterations of organelle dynamics. A** Nucleoli/nuclei size-ratio per cell over time (#of quantified nuclei in light-gray), **B** Lipid droplet dry mass per cell (#of quantified single cells in light-gray), **C** Mitochondria dry mass density (#of quantified mitochondria in light-gray), **D** Mean mitochondria length per cell (#of quantified cells in light-gray), **E** the ratio per cell of lipid droplets <400 nm away from the nearest mitochondria (#of quantified cells in light-gray). Violin plot representation of non-infected, Wuhan-,

Omicron-infected, and Syncytin-1-expressing cells. Each violin plot represents the distribution of the segmented single cells or organelles contained within the indicated period. Bin-to-bin two-sided *t*-tests *p*-values are indicated below each experiment. Single cells and organelles studied in each of the mock, Wuhan, Omicron, and Syncytin-1 coming from at least three different movies (see Supplementary Movies 1–19).

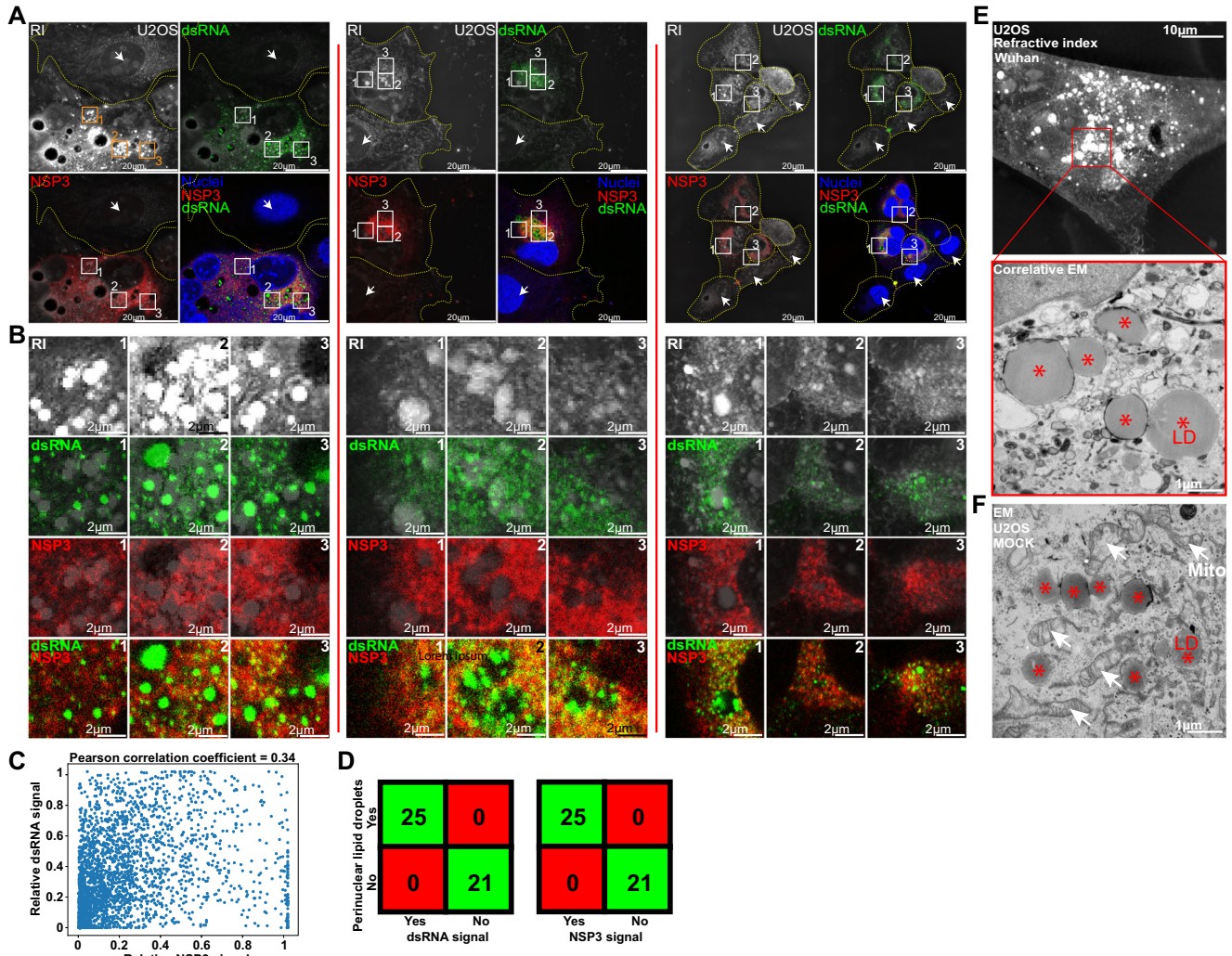

**Fig. 5 | Perinuclear lipid droplet accumulation is a marker of infection. A** and **B** Comparison of the refractive index (RI) signal acquired using holotomographic microscopy (HTM) with the double-stranded (ds) RNA and NSP3 immuno-fluorescent signals acquired with confocal microscopy. dsRNA spots are detected over and around lipid accumulations while the homogenous NSP3 signal is excluded from them (# of independent acquisitions = 8). **C** Scatter plot of NSP3 against dsRNA fluorescent signal and Pearson correlation coefficient of 0.34 confirm the absence of signal colocalization. **D** Confusion matrix. Cells with perinuclear lipid

droplet accumulation have SARS-CoV-2-induced dsRNA and NSP3 signals. Cells with no LD (arrow) show no dsRNA and NSP3 signals (see Supplementary Fig. 8). **E** Correlative holotomography/electron microscopy (EM) in infected cells. No mitochondria surround infection-induced lipid droplets (# of independent acquisitions = 5) (see also Supplementary Fig. 4). **F** Electron microscopy in non-infected cells, where LD is close to mitochondria (red stars: LD white arrows: mitochondria, # of independent acquisitions = 3).

our observations (Figs. 4E, 5D and E, Supplementary Figs. 8, 9) that SARS-CoV-2 lead to the spatial separation of LDs and mitochondria and suggesting a profound alteration of LDs metabolism. The infection-induced OCR network (Fig. 6B) was mostly altered around LDs, with minor to no changes elsewhere. Thus, our approach captures organelle connections. occurring during SARS-CoV-2 infection: LDs are specifically perturbed; the syncytium state is linked to LDs, and the OCR structure points towards LDs through the cytosol node (Fig. 6B).

The OCR of Syncytin-1-expressing cells (Fig. 6C) helps in understanding the relative roles of cell fusion versus viral infection.

In Syncytin-expressing cells, the syncytial state had no direct impact on one specific organelle and was only loosely related to the rest of the network through the cytosol (Fig. 6C). The formation of a syncytium is a complex, broad phenomenon that implies more than the fusion of plasma membranes on which we rely to identify it in our dataset. The molecular mechanisms at play, including cytoskeleton reorganization, nucleus clustering, and rapid mixing of different trafficking, signaling, and metabolic systems, are expected to have broad

consequences that will not be captured as directional dependencies but rather be seen in the way organelles are wired together. The core OCR linking cytosol, mitochondria, and the nuclei-nucleoli duo remains, confirmed the robustness of our approach. In these syncytia, the link between LDs and the rest of the OCR was no longer visible. Thus, one can speculate that one role of the syncytium is to change how organelles interact together. The virus may thus trigger the syncytium to establish a favorable OCR system to facilitate its spread.

Our bootstrapped BN analysis (Fig. 6B) suggested a direct link between the formation of the infection-induced syncytium and LDs. To further validate our systems approach, we performed a high-frequency HTM time-lapse experiment with one image being recorded every 2 min (Fig. 7A). The early moments of syncytium formation were generally followed by LD growth While infectious and non-infectious syncytia looked globally similar, the non-infectious syncytia did not promote LD growth on its own (Figs. 3D and 4B).

We finally used confocal microscopy of tubulin immuno-staining as a complementary approach to further investigate potential

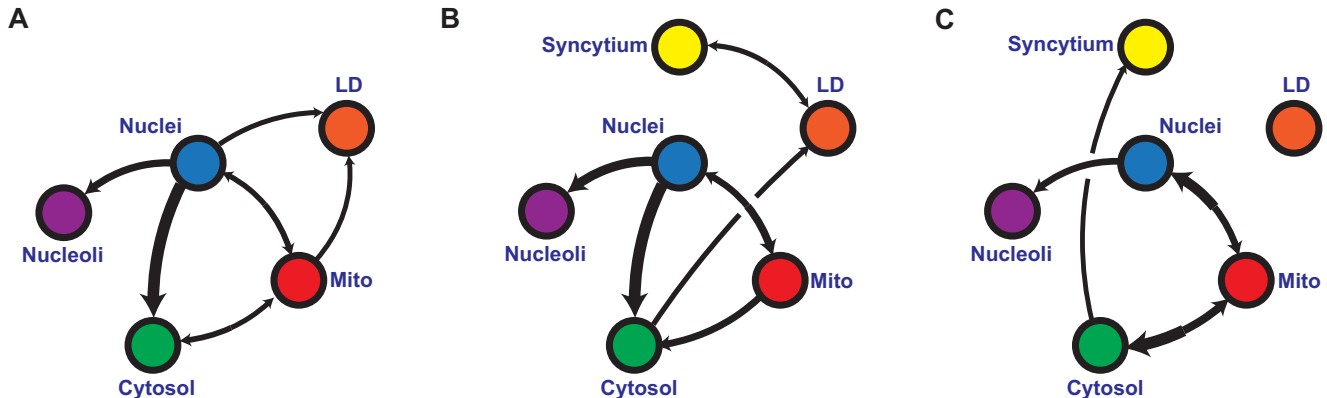

**Fig. 6 | Bootstrapped Bayesian networks indicate that SARS-CoV-2 infection changes organelle cross-regulation. A–C** Averaged Bayesian networks representing the dependencies (equivalence classes) between mitochondria, nuclei, nucleoli, lipid droplets, and cytosol dry mass states and the syncytial state of **A** non-infected cells, **B** infected cells, and **C** Syncytin-1-expressing cells over 10,000 randomly resampled datasets. All edges at each iteration were validated by two-sided chi-squared, g-squared, log-likelihood, and Cressie-Read pairwise tests, all displaying a *p*-value < $10^{-6}$ over discretized and non-discretized data (see Supplementary Tables 1–4 for two-sided chi-squared test details) (# of independent time-lapse acquisitions for each condition ≥ 4).

**Fig. 7 | SARS-CoV-2-induced syncytia displays a flat radial tubulin signal and accelerates lipid droplet growth. A** High-frequency time-lapse imaging of U2OS cells infected by SARS-CoV-2 Wuhan strain (see also Supplementary Movie 20). **B, C** Representative images of confocal immunofluorescence microscopy in SARS-CoV-2 Wuhan- or Syncytin-1-induced syncytia. Green: Tubulin, Blue: DAPI. Pink and yellow arrows are the sections along which signals plotted in **D** are sampled. **D** Representative tubulin signal profiles from a nuclei cluster towards cell boundaries. **E** Occurrence of a flat radial actin pattern.

differences between the two types of syncytia. Compared to Syncytin-1 expressing cells, SARS-CoV-2-induced syncytia displayed a flat microtubule signal profile (Fig. 7B and C) from the clustered nuclei towards the syncytium boundaries (Fig. 7D). This flat microtubule signal distribution was detected in 77% of the infected syncytia and only in 20% of Syncytin-induced fused cells (Fig. 7E). In the 80% remaining cells, a gradient was seen, with more tubulin surrounding the nuclei than in the cell boundaries. Thus, in addition to the events captured by

HTM, the cytoskeleton is differentially modified in SARS-CoV-2-infected or Syncytin1-induced syncytia. These differences could modulate LD accumulation and mitochondrial redistribution, opening future investigative directions.

## Discussion

We demonstrate here that AI-enhanced label-free microscopy holds great promise for the investigation of pathological processes such as virus infection and offers possibilities to quantitative cell biology in general[61]. The emergence of high-content fluorescence microscopy[62] and massively multiplexed labeling methods[63] revolutionized our understanding of cells' systems[64], whether they are unperturbed, adapting to changing micro-environments[65], or reacting to drug treatments[66]. Yet, these methods perturb living cells and are not always suited for monitoring fine biological dynamics over long periods of time.

Among currently available label-free techniques, holotomographic microscopy (HTM) produces images whose quality enables effective computer vision solutions[20,67]. To achieve this aim, it is important to overcome the laborious task of ground truth generation[68,69] and to choose the adequate object detection technique, where deep learning is not always the most effective solution. The ground truth that we produced allowed for a U-NET segmentation of simple, large objects like cells, nuclei, or nucleoli, but was poorly suited for the detection of mitochondria that are sparser and thinner objects. A more explicit control of the receptive field in the case of the pixel classification approach, which depends on the blurring steps applied to images before feature extraction, was more adapted for objects spanning only a few pixels only. our work is thus particularly relevant to the field of quantitative mitochondrial biology[37,70,71], given the susceptibility of that organelle to phototoxic stress and label-induced perturbations[8,72,73].

Our object detection strategy transforms the limitation of HTM, which is the complex nature of the images it provides, into an advantage: having access to biologically relevant, direct, simultaneous measurements of organelles within single cells. This opens a vast landscape of possibilities for system investigations. Such a dataset is particularity adapted for causal investigations due to the reasonable number of dependencies to evaluate (Supplementary Fig. 10). We could thus define in this study organelle cross-regulation (OCR) networks. There are pending questions: will the same OCR network topologies be observed in other cell types or in response to other viruses? Future work will help determine whether we have uncovered conserved regulation networks for synchronizing organelles operating in different cellular conditions.

There are limitations to our study. Firstly, Bayesian networks are directed acyclic graphs and, as such, cannot be circular or contain feedback loops, which are natural components of many biological systems. Thus, the use of causal inference provides a partial view of OCR networks and of their potential rewiring upon perturbations. Targeted metabolic or genetic perturbations to challenge the hypothesis gathered through network inference will help further characterize OCR dependencies. Secondly, key subcellular structures such as the Golgi or the endoplasmic reticulum are not captured by the current sensitivity of HTM and are therefore absent from our analysis. HTM could be associated with other label-free[74], fluorescence, or electron microscopy in correlative approaches. Such association is essential to advance both label-free and label-based imaging worlds. This is illustrated here by the assessment of mitochondrial localization by CHEM, the colocalization of LD, dsRNA, and NSP3 by correlative HTM/confocal microscopy, or the characterization of tubulin network modifications during SARS-CoV-2 infection and syncytia formation. Thirdly, we used only one cell type, U2OS cells expressing the ACE2 receptor, to establish the first set of discoveries in an uncharted territory. It will be important to extend such analysis to other cell types,

for instance, those that do not or poorly form syncytia upon infection. The use of viral strains carrying GFP-tagged viral proteins or other markers will facilitate the identification of infected cells[75]. This will allow us to further explore the whole infection process and better characterize the isolated role of syncytia during infection such as the impact of almost instantaneous small molecules redistribution and the mixing of multiple signaling or metabolic states. Fourthly, to ensure capturing all biological structures despite dynamic variations in z distribution, we work with a typical 2D projection procedure that is applied similarly to all 3D sample reconstructions (Supplementary Fig. 11a, b). Thus, our quantifications approach the true dry mass content rather than being absolute (Supplementary Fig. 11c). It is important to mention that a full 3D-based quantification is not an absolute reference either, even with the best possible microscopic device, absolute quantitative knowledge is out of reach. This is why controlled experiments like those made in our study, are and will remain essential. Importantly, a 3D-based quantitative investigation at a similar experimental scale represents a massive technical and computational challenge that is a matter of future studies. Finally, our work quantitatively describes the multiple cellular events associated with SARS-CoV-2 infection in real-time. An analysis of cells expressing individual viral proteins, such as, for instance, ORF9b[76], that interacts with the mitochondrial protein TOMM70[77], or NSP6, that mediates contact between DMVs and LDs[78], will provide clues on the impact of SARS-CoV-2 on the dynamics and shape of mitochondria.

In summary, we developed an approach combining HTM, AI-assisted analysis, and causality inferences using Bayesian network modeling, to assess the impact of SARS-CoV-2 on the dynamics of cellular organelles and OCR. We report that the virus directly alters LDs, mitochondria, nuclei, and nucleoli. We describe how those organelles can influence each other. We also propose that the infectious syncytium is likely favoring a pro-virus cellular environment. This approach opens exciting possibilities to analyze any pathogen, drug effects, and physiological or pathological events affecting cell life, including nutrient variations, metabolic adaptation, and malignant transformation. It holds promise to lead to insights into the dynamics of a vast range of biological processes.

## Methods

### Holotomographic microscopy time-lapse acquisitions

All label-free images were acquired using a 3D-Cell Explorer-fluo (Nanolive SA, Tolochenaz, Switzerland) microscope. This microscope is equipped with a 520 nm laser for tomographic phase microscopy; the irradiance of the laser is $0.2\,nW/\mu m^2$, and the acquisition time per frame is 45 ms. It is equipped with a Blaser ace acA1920-155um camera, and an air objective lens (NA = 0.8, magnification ×60). The microscope is equipped with a fluorescent module (pE-300ultra, CoolLED) for standard epifluorescence images of the sample in three different channels: Cy5 (excitation peak 635 nm), TritC (excitation peak 554 nm) and FitC (excitation peak 474 nm). The microscope is equipped for long-term live cell imaging: temperature, humidity, and gas composition. The incubator chamber (Okolab) keeps the sample at 37 °C, is closed by a heating glass lid to prevent condensation, and is connected to a gas mixer (2GF-Mixer, Okolab) to maintain 5% of $CO_2$. The humidity module ensures a 90% relative humidity within the chamber. The 520 nm laser is divided into two beams to create an interferometer setup[79]. One of the beams, referred to as the object beam, interacts with the sample before being gathered by the 60× objective lens. Simultaneously, the other beam, left unperturbed, functions as the reference beam. When meeting on a dichroic mirror, the two beams create an interference pattern a.k.a. a hologram, further captured by a CMOS camera (Supplementary Fig. 1a). This conventional holographic technique is augmented with the rotational scanning of the sample[80] (Supplementary Fig. 1b) using a rotating mirror that reduces the problem of coherent diffraction noise[81] to imperceptible levels

(Fig. 1B and C). The series of holograms are then assembled by complex field deconvolution[82,83]. Our HTM device incorporates dynamically adjustable mirrors that enable its optics to self-regulate throughout the acquisition and accommodate variations in the sample, such as medium evaporation or variations in meniscus curvature. Combined with an automated stage and a laser based autofocus, our HTM device has stable performances over long periods of time. Finally, the HTM device possesses an epifluorescence module allowing for simultaneous fluorescence and RI imaging (Supplementary Fig. 1c).

For imaging, 50,000 cells were plated in a 35 mm No.1.5 ibidi polymer coverslip bottom dish. After 24 h, the media was replaced with fresh media containing the indicated dose of the virus. Dish were then placed in the incubator chamber of the microscope and imaging was started 3 h post-infection.

### Live fluorescent controls

For mitochondria labeling, TMRE was added to the media for 30 min, before washing and cell imaging. For siRNA experiments, Dharmacon smartpool siRNA directed against human-OPA1 siRNA (Dharmacon SmartPool M-005273-00-0005) or a control siRNA (Dharmacon SmartPool D-001210-02-05) was used. For nucleoli labeling, cells were transfected using lipo3000, as described by the manufacturer, with GFP-GRI, a subunit of the Simian Foamy virus that contains a nucleolar localization signal fused with GFP[84]. 24 h post-transfection cells were imaged using the Nanolive. For lipid droplet labeling, Bodipy 493/503 was added in serum-free media for 15 min, before washing and imaging.

### Immunofluorescent labeling

Infected cells were fixed with 4% PFA for 30 min. For correlation with Nanolive microscopy, the dish was scratched to provide a visual landmark. They were then imaged on the Nanolive. They were permeabilized with 1% Triton for 10 min and blocked overnight with PBS/1% BSA/0.05% sodium azide. Mouse anti-dsRNA J2 (1:100, RNT-SCI-10010200; Jena Bioscience), sheep anti-SARS-CoV-2 nsp3 (1:200, DU67768, MRC PPU) were used overnight in blocking buffer. Donkey anti-sheep 488 (#A-11015, Thermo Fischer Scientific) and Donkey anti-mouse 647 (#A-31571, Thermo Fischer Scientific) were used at a 1:500 dilution in blocking buffer for 1 h. Imaging was performed using a Leica SP8 confocal microscope. For cytoskeleton imaging, infection was performed in a 96-well plate (Perkin Elmer). Infected cells were fixed with 4% PFA for 30 minutes, permeabilized with 1% Triton for 10 min, and blocked overnight with PBS/1% BSA/0.05% sodium azide. Mouse anti-tubulin (1:100, ProteinTech 66031-1-Ig) was used overnight in the blocking buffer. Goat anti-mouse 488 (#A-11015, Thermo Fischer Scientific, # A-11001) was used at a 1:500 dilution in a blocking buffer for 1 h. Imaging was performed using the Operetta Phenix (Perkin Elmer).

### Electron microscopy

50'000 cells were plated in a MatTek 35 mm Dish (P35G-1.5-14-C-GRD). The next day, the media was replaced with fresh media containing the indicated dose of the virus. After 24 h, they were fixed with 4% PFA for 30 min and imaged using holotomographic microscopy. Cells were then fixed in 2.5% glutaraldehyde (Sigma) in 1X PHEMS buffer overnight at 4 °C. Samples were washed 3 times in 1X PHEM buffer and post-fixed in 2% OsO$_4$ (Electron Microscopy Sciences) + 1.5% potassium ferrocyanide in water 1 h in the dark. After three washes in water, they were incubated for 20 min in 1% uranyl acetate in ethanol 25%. Samples were gradually dehydrated in an ethanol series (50%, 75%, 95%, 3 × 100%) and then embedded in EMbed-812 epoxy resin (Electron Microscopy Sciences), followed by polymerization for 48 h at 60 °C. Thin sections (70 nm) of the region of interest were cut with a Leica Ultramicrotome Ulracut UCT stained with uranyl acetate and lead citrate. Images were acquired using a Tecnai T12 120 kV (Thermo Fisher) with a bottom-mounted EAGLE camera.

### Cells

HeLa, 3T3-derived preadipocytes, HEPG2, and U2OS cells were purchased from ATCC. U2OS cells were stably transduced with pLenti6-human-ACE2 as previously described[29] or with pCW57.1_Syn1. The doxycycline-inducible PCW57.1_Syn1 plasmid was obtained by cloning the previously described phCMV_Syn1[85] into pCW57.1 (Addgene) using the NheI + AgeI restriction for both plasmids. U2OS cells stably expressing PCW57.1_Syn1 were generated by lentiviral transduction and puromycin selection. Synyctin-1 expression was triggered by treatment with doxycycline. All cells were cultured in DMEM with 10% fetal bovine serum (FBS), 1% penicillin/streptomycin (PS). For imaging and infections, 25 nM HEPES was added to the media. 10 µg/ml blasticidin was added to U2OS cell cultures. Cells were routinely screened for mycoplasma.

### Virus

The reference Wuhan strain and the Omicron BA.1 variant have been described previously[86,87]. The viruses were isolated from nasal swabs of two patients using Vero E6 cells and amplified by one or two passages. Ethical regulations were followed. Both patients provided informed consent for the use of the biological materials. Titration of viral stocks was performed on Vero E6 cells, with a limiting dilution technique enabling the calculation of the median tissue culture infectious dose, or on S-Fuse cells[88].

### Holotomographic image processing

All the time points composing the time-lapse imaging experiments were reconstructed and stored as 3D volumes by Eve, the software that controls the 3D cell explorer microscope and early data management. Custom routines were used to convert 3D volumes into 2D maximal projections along the z-axis from −2 to +6 µm around the point of focus, which ensures always capture of entire cells and their organelles while those biological objects may move along the z-axis.

### Deep Learning object detection

A custom U-Net architecture (Supplementary Fig. 2, model available on https://github.com/SlowProspero/SARS_AI/) was used to segment nuclei, nucleoli, and cells.

The U-Net consists of an encoder-decoder structure meeting at a bottleneck layer. The encoder component has four blocks, each comprising three 3 × 3 convolutional layers followed by rectified linear unit (ReLU) activation and batch normalization. Between these blocks, 2 × 2 max-pooling layers are used to down-sample the feature maps, and dropout layers of 0.5 are used to prevent overfitting. The input layer is the full 448 × 448 image. Because of the 3 × 3 convolutional layer, and 2 × 2 max pooling layer, the following blocks are then 224 × 224 pixels and 32 feature maps, 112 × 112 and 64 feature maps, 56 × 56 and 128 feature maps, 28 × 28 and 256 feature maps.

The decoder component is symmetric to the encoder, it consists of four blocks each like the encoder blocks. However, instead of max-pooling layers, transposed convolutional layers (also known as deconvolutional layers) are utilized to up-sample the feature maps. The first decoder block is in common with the last encoder block. Skip connections are established by concatenating the feature maps from the corresponding encoder part to its symmetrical decoder part. Finally, the size of the bottleneck layer is 28 × 28 and 512 feature maps.

Predictions in images larger than 448 × 448 involve making patches of prediction of size 448 × 448. Such window is then moved across the image stepwise 50 by 50 pixels, horizontally and vertically, and the overlapping predictions are averaged.

Training processes were run in a tensor-flow 2.10.1/Python 3.8.10 environment for 500 epochs for the nuclei and nucleoli models on an Nvidia RTX3060 GPU with 12 GB of VRAM and for 50 epochs for the cell models on an Nvidia RTX A4000 16GB of VRAM. The ground truths for the cells, nuclei, and nucleoli, were made semi-manually with the

help of a custom labeling tool and the guidance of fluorescently stained cells. No ground truth was generated for the cytosol compartment as this mask type is generated from subtracting organelles masks from the cell mask, as described below in the dedicated section.

### Nucleus
Our nucleoli model was trained with a dataset of 655 holotomographic microscopy images of mammalian cells images randomly split into a training set (589 images) and a testing set (66 images). These images come from 58 acquisitions and include multiple cell lines. Each image contains from 1 to 23 nuclei; all have a size of 480 × 480 pixels achieved by cropping or zero-padding. Probability maps provided by the model are binarized with a 50% probability threshold. All objects smaller than 10 pixels are rejected. The training took 4 h and 19 min.

### Nucleoli
Our nucleoli model was trained with a dataset of 495 holotomographic microscopy images of mammalian cells randomly split into a training set (445 images) and a testing set (50 images). Each image contains at least a dozen nucleoli and has a size of 480 × 480 pixels achieved by cropping or zero-padding. Probability maps provided by the model are binarized with a 50% probability threshold. All objects smaller than 10 pixels are rejected. Potential unspecific signal outside the nucleus is removed thanks to the nucleus masks predicted by our nucleus model. The training took 4 h and 13 min.

### Cells segmentation
The first part of our cell segmentation process aims at obtaining rough cell segmentation without precise cell border detection. This U-Net model was trained with a dataset of 1445 holotomographic microscopy images of mammalian cells randomly split into a training set (1295 images) and a testing set (150 images). Each image contains from 1 to 10 cells and has a size of 480 × 480 pixels achieved by cropping or zero-padding. Probability maps provided by the model are binarized with a 50% probability threshold. All objects smaller than 10 pixels are rejected. The training took 51 min. The second part of our cell segmentation process takes the U-Net produced cell blobs binary masks as a seed for a precise cell edges detection using a propagation algorithm approach.

### Ensemble pixel classification for mitochondria detection
Mitochondria were segmented using a tailored pixel classification approach inspired by previous seminal work[32]. Our code can be found at https://github.com/SlowProspero/SARS_AI/. Because the simple refractive index value is not a reliable organelle signature, we increased the pixel feature space dimensionality by applying on refractive index images a set of convolution filters using the VIGRA Computer Vision Library (v1.11.1) for a total of 21 dimensions. Thus, each pixel is described by its respective refractive index value and by its value in the 20 convolved images created by applying the following 10 filters on the refractive index image at two different sigma values (1.4 and 2.0): gaussian smoothing, difference of gaussian, gaussian gradient, gaussian gradient magnitude, hessian of gaussian, hessian of gaussian eigenvalue, Laplacian of gaussian, structure tensor eigenvalue, tensor determinant, and tensor trace. The probability for each augmented pixel of our images to be part of a mitochondria or lipid droplet was then evaluated by an extra-tree classifier (scikit-learn v1.2.2) diverging from the default hyperparameter setup only by the number of estimators that is equal to 200. Each resulting probability image is then transformed into a binary mask using an adaptive background removal approach (OpenCV, cv2.threshold). Our mitochondria extra-tree classifier was trained thanks to the labeling of 149 images from two different cell lines (CHO and Preadipocytes) coming from 14 different acquisitions.

### Detection of lipid droplets
Lipid droplets were automatically segmented using Nanolive lipid droplet assay software that identifies spherical structures of even signal distribution over specific local refractive index maxima. The produced lipid droplet segmentations were further used for custom metrics calculations.

### Definition of the cytosol compartment
The cytosol compartment is defined as the mask created by the subtraction of all organelles masks (Nucleus, nucleoli, lipid droplets, mitochondria) from the cell segmentation mask.

### Metrics calculation
The instantiated masks of cells, mitochondria, lipid droplets, nuclei, and nucleoli and cytosol were used to extract the spatial (shape and size descriptors, centroid position) and RI-derived (dry mass, textures) features of each of the mentioned biological objects in each frame of each time-lapse experiment (Supplementary Movies 1–19). This was performed using a scientific Python environment and the library scikit-image. Each object's dry mass content was calculated from its refractive index value using a linear calibration model[32]. The data were exported as a.csv file into a Python environment and plotted with the Python library matplotlib for figure making or used through the bnlearn python library to establish our organelle cross-regulation networks.

### Bayesian network structure learning
We performed our Bayesian network (BN) structure learning using the bnlearn[56] python package, to determine which BN captures the best the directional dependencies that exist between our dataset variables, lipid droplets, mitochondria, nucleus, nucleoli, and cytosol dry mass densities, as well as the nature of the single cell, normal or syncytium. This was performed 10,000 times in a typical bootstrapping approach where each of the 10,000 datasets has the same size as the original dataset and is generated by random sampling with replacement. This approach allows the capturing of the most robust network structure through BN averaging and diminishes the impact of potential experimental and time-related technical biases.

The best BN for each condition over each dataset was defined using greedy score-based structure learning. For this study, we used the K2 and hill climb search-and-score algorithms, which incrementally test BN alternatives to improve the default scoring function BIC, which itself determines the probability of the BN structure given its training data.

We performed multiple independence tests to define the edge strengths of our BN using each of the three models and their related discretized and non-discretized data. For each pair of our learned BN, a chi-squared, g-squared, loglikelihood, and Cressie–Read statistical test is performed to determine its p-value and associated score. The mean chi squared scores over all bootstrapped iterations are then used as a quantification of the strength of the tested edges. Only edges satisfying the pairwise statistical test and observed at a frequency superior to 0.15 are displayed (see Supplementary Tables 1–4).

### Reporting summary
Further information on research design is available in the Nature Portfolio Reporting Summary linked to this article.

## Data availability
Source data are provided with this paper. All data supporting the findings of this study are available within the paper and its supplementary Information. Further information can be requested from the lead contact at mathieu.frechin@nanolive.ch. Source data are provided with this paper.

## Code availability

The Bayesian network code is available at https://github.com/SlowProspero/SARS_AI/.

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

## Acknowledgements

We thank Timothée Bruel, Nicoletta Casartelli, and Sasha Legrosdidier for critical reading of the manuscript, members of the Virus and Immunity Unit for discussions and help, Perrine Bomme and the Ultra-structural Bioimaging Unit (UBI) and UtechS Photonic BioImaging (UPBI) core facilities (Institut Pasteur). We thank Kyle Van Der Langemheen for establishing Optuna. N.S. is funded by the ministère de l'Enseignement supérieur et de la Recherche. O.S. laboratory is funded by Institut Pasteur, Fondation pour la Recherche Médicale (FRM), ANRS-MIE, the Vaccine Research Institute (ANR-10-LABX-77), HERA European program (DURABLE consortium), Labex IBEID (ANR-10-LABX-62-IBEID), ANR/FRM Flash Covid PROTEO-SARS-CoV-2 and IDISCOVR. We are grateful for support for equipment from the French Government Program Investissements d'Avenir France BioImaging (FBI, No. ANR-10-INSB-04-01) and the French gouvernement (Agence Nationale de la Recherche) Investissement d'Avenir program, Laboratoire d'Excellence "Integrative Biology of Emerging Infectious Diseases" (ANR-10-LABX-62-IBEID). We thank Thibault Corteoux and Lisa Polaro for connecting Nanolive's Deep Quantitative Biology team with the Institut Pasteur. We thank all Nanolive's employees for their relentless appreciation, encouragement, and support.

## Author contributions

Conceptualization, N.S., B.M., T.W., O.S., and M.F.; Methodology, N.S., B.M., N.C., L.A., H.M., A.J., J.B., A.M., O.S., and M.F.; Data curation, L.A., H.M., A.J., A.M., and M.F.; Software, L.A., H.M., A.J., A.M., and M.F.; Investigation, N.S., N.C., B.M., and M.F.; Visualization, N.S. and M.F.; Writing—original draft, M.F; Writing—review & editing, N.S., T.W., O.S., and M.F.; Funding acquisition, O.S.; Resources, O.S. and M.F.; Supervision, T.W., O.S., and M.F.

## Competing interests

L.A., H.M., A.J., A.M., and M.F. are employees of Nanolive SA. The remaining authors declare no competing interests.
