## [Peer Review File · Nature Communications]

Reviewers' Comments:

Reviewer #1:

Remarks to the Author:

I have been asked to concentrate in my review on the Bayesian network (BN) modeling aspects of this study, as it pertains to my area of expertise. Therefore, I will not comment on the overall significance/impact in the broader SARS-CoV-2 infection context. That being said, I agree with the authors that using BNs for dissecting cell organelle regulation patterns is a novel and potentially fruitful idea.

The BN modeling application presented in the manuscript is sound overall. However, there are caveats, some major. Below I list the issues that need to be resolved before the paper is accepted for publication:

1. The authors use the term "Bayesian statistics" throughout the manuscript. This is misleading, as BNs have little (if anything) to do with Bayesian statistics. BN is a probabilistic graphical model, part of the machine learning (and not Bayesian statistics) toolkit. I suggest the authors use "Bayesian network modeling" instead of "Bayesian statistics".
2. In "Bayesian network structure learning" section (lines 1018+), there is a number of issues. First, there is no "causal" learning in a strict sense. The authors should tone down the language. BNs establish directional dependencies that might suggest directional causality, but there is no causality analysis (or intervention experiments with subsequent BN re-construction) in the study. Second, K2 is not a scoring function. It is a search-and-score algorithm. Third, what was the actual scoring function used? The default scoring function in bnlearn is BIC, but the authors need to specify that explicitly.
3. Line 396. It's conditional dependencies, not relationships.
4. Why use bootstrapping? This can be very misleading with BNs because of the multicollinearity (that BNs are designed to resolve in the first place). "Averaging" the resolutions of, say, a particular V-structure in a BN over 10,000 replicates strikes me as more potentially confusing than illuminating. I believe bootstrapping is, at best, unnecessary in this context, especially given the favorable dimensionality of the data. Bootstrapping will quantify robustness, but not the actual "signal strength".
5. Following up the direct dependencies established via BN with pairwise statistical tests (chi2, in this case) is a good idea, and inspires confidence. However, was it done over the discretized variables? I assume Yes. In which case, I would suggest using different statistical tests (e.g., a logistic regression or a point-biserial, for continuous/discrete mixture) without discretization.
6. Lines 401, 451. Again, "causal" is too strong.
7. (MAJOR) In line 422, the authors talk about the network's "rewiring". However, all the "rewiring" we can see is a slightly different factorization of the joint probability distribution. This could happen for any number of reasons, such as the glaringly obvious one of introducing a new node/variable (Syncytium) into the graph. Therefore, the "rewiring" conclusions (lines 425, 458) are not strongly supported. I am not saying that BNs are not useful at suggesting changes --- they are, but I'd rather use the language of "suggesting a direct link" than "identifying a direct link".
8. (MAJOR) How did the authors discretize the variables? BNs are notoriously sensitive to discretization. A summary table (variables, their descriptive statistics and distributions/shapes, discretization bins) should be present in Supplementals.
9. The authors are talking about DAGs (directed acyclic graphs), but the graphs in Figure 6 look more like equivalence classes (with many bidirectional edges). This is fine, but should be mentioned in Figure 6 legend to avoid the readers' confusion.
10. How are the edge strengths in Figure 6 determined, via bootstrap? This should be in Figure 6 legend. bnlearn has many options for edge strength, some more appropriate here than the others.

I'd rather the authors use scoring criterion-based edge strengths than bootstrapping (see Comment 4 above).

In general, BNs are useful for getting a holistic view of the domain, and for hypothesis generation, but not so much for comparative hypothesis testing, as the authors are doing here with the "rewiring". I don't think that the "rewiring" analysis is convincing in itself (although the subsequent experiments by the authors do inspire some measure of confidence).

To illustrate, when two edges are replaced by one different edge (such as in Figure 6 A->B transition), the correct "apples-to-apples" analysis would be (i) to skip the bootstrap, (ii) do the pairwise statistical tests between **all** pairs (including Cytosol-LD in Figure 6A and Nuclei-LD and Mito-LD in Figure 6B), not just along the direct dependencies suggested by the BNs, and (iii) compare the p-values for all (direct and transitive) pairwise relationships in Figure 6AB networks. Then, one would have a stronger argument in support of rewiring.

Reviewer #2:

Remarks to the Author:

In the present article, authors highlights the subcellular restructuring triggered by SARS-CoV-2 infection introduces alteration in cell population dynamics and cross-talk between compositions which can be studied by a fresh AI-enhanced, label-free approach for real-time examination. With meticulous attention, they documented the progression and scope of these changes in live cells, by capturing and measuring cellular and organelle dynamics in real-time, along with providing extensive evidence of the virus's direct impact on lipid droplets, mitochondria, and nuclei. The manuscript can have potential interest for the readers who plan to work in a similar field. References are updated and correctly cited in the manuscript. However, I want to express some concerns which are needed to be addressed in order to improve the scientific quality of the manuscript as well as to be considered for publication.

The page and line mentioned in the comments is on the basis of main article pdf file provided.

Major points

1. The claim "However, we detected 142 similar mitochondrial movements in both SARS-CoV-2- and Syncytin-1-induced syncytia." Situated at line 145 page 5 Is not evidently depicted in the mentioned figure (fig.1). How is this "mitochondrial movements" measured here in fig.1?
2. I don't find "Fig.1 (D) Time lapse imaging data projection for 2D computer-118 vision analysis and qualitative assessment of cells." In the figure embedded in article pdf.
3. The claim in line 288-290 page 14 that "LD remained well-defined even when their size increased or when they moved from the cytosol to the perinuclear region (Fig. 3C and 3D)." is not much decipherable from figure 3C-D where nuclear staining is not described.
4. Can authors discuss with probable molecular mechanism how "SARS-CoV-2 could use syncytia formation to promote mitochondrial alterations " proposed in line 322 in page 14?
5. As there are many cellular dsRNA intermediate exists, author should clearly discuss how the specificity is ensured in the treatment mentioned in , "Anti-dsRNA antibody selectively recognizes viral RNAs and not cellular RNAs."(line 351, page 16).
6. A correlation scatter (e.g., pearson) plot for the collocalisation claim of "dsRNA and NSP3 "in line 353,354 page 16 would be more consolidating.

Minor points

1. Methodological description of techniques or protocols not being reported for the first time in this paper, e.g., HTM should be a part of "Methods" but in this case has been provided in "Results" section (pdf page 2 line 84-90, 93-97). Authors should consider reducing the detailing in result part until it is profoundly indispensable to describe the results and the settings are different in every observations or events.

Overall the present study exhibits descriptive observational narration exploiting number of online analysis tools which proposes utilizable avenues of machine learning and statistical models in

finding intracellular organelle cross talk and response against a stress. They made good use of Bayesian networks (BN) to model the hierarchy and regulation existing between cell organelles which occurred to me quite novel and except of some mention regarding some nuclear bodies in the following paper (A Bayesian network model of proteins' association with promyelocytic leukemia (PML) nuclear bodies By Mikael Bodén et al. PMID: 20426694) no other mention of such kind is easily available.

Reviewer #3:

Remarks to the Author:

Review of Dynamic label-free analysis of SARS-CoV-2 infection reveals virus-induced subcellular remodeling.

The manuscript submitted by Saunders and co-authors that proposes a methodology to analyse images of SARS-CoV-2. The manuscript has interesting contributions in the acquisition, with label-free holo-tomographic microscopy, a deep-learning based segmentation of the images, and a series of observations derived from the segmented datasets. My expertise in the acquisition and biology is limited, thus I will concentrate on the processing of the data. Still, I am well impressed by the acquisition of the images, the quality is very good and provide excellent resolution for the subsequent segmentation and analysis. In terms of the comparisons requested, these are well documented with a large number of images to compare with.

There is a criticism of lack of novelty in the ML. I would not consider that training a well-known UNet for a specific application would count as novelty. Indeed, there have been many newer architectures based on UNet that improve times or results on certain datasets. Contribution in the quality of detection would require a numerical comparison with other methodologies. However, I would not think that it is necessary to have a contribution with a new architecture for this paper that has already provided novelty in other areas, namely the acquisition and the findings from the data itself. Thus, if UNet provides a sufficiently good segmentation, that is more than enough in this paper. I would just highlight, that the section "Deep Learning object detection" (lines 947 onwards) should provide a better description of the architecture, number of layers, number of images with which it was trained were there patches or whole images, time to train, software used (somewhere else Python is mentioned but not in the main document).

Reviewer #4:

Review of the manuscript:

Dynamic label-free analysis of SARS-CoV-2 infection reveals virus-induced subcellular remodeling, by Saunders *et al.*

Summary of contributions:

Saunders. *et. al* report dynamic measurements of organelle remodeling in U2OS-ACE2 cells during SARS-CoV-2 infection using label-free holo-tomography (HTM). The changes in organelle morphology and composition are measured in terms of the dry mass. They integrate multiple machine learning and deep learning algorithms (random forest classifier, convolutional neural networks, and Bayesian modeling) to identify how key organelles (mitochondria, lipid droplets, nuclei, nucleoli) are remodeled during infection and suggest some causal links between phenotypes. Segmentation of organelles is guided by fluorescent markers. Some of the key findings, such as increase in the size and density of lipid droplets, are validated with complimentary electron microscopy and confocal microscopy.

Overall, above contributions report exciting advance at the interface of live cell imaging, machine learning, and infection biology. The paper also clarifies some of the strengths and weaknesses of the approach, e.g., organelles that have poor contrast in refractive index cannot be profiled with HTM, the segmentation models they report are only for 2D images, and Bayesian model they employ cannot capture feedback loops. Writing is clear and well structured.

We found following areas of improvement in clarity and accuracy:

- HTM is a 3D imaging system. Authors should report z flythroughs from the apical to basal side of the cells in the infected and uninfected conditions and compare them with 2D maximum projection. What is the effect of 2D maximum projection on the estimate of the dry mass? We'd argue that data in Fig. 4B and 4D cannot be interpreted in the units of pg or pg/um², since the measurements are done from the maximum projection rather than actual reconstruction. For example, what is the percentage error in dry mass measurement of nuclei due to endoplasmic reticulum on top of it which is not distinguished.
- How the cytosol was segmented requires an elaboration: The authors explain that a sharpening of the outlines was performed by object propagation within the RI signal for cytosol segmentation.
 - Validation of cytosol segmentation same as other segmented organelles is not presented. What is the ground truth in this case? Is it generated manually by experts?
 - How well does this algorithm work as authors track formation of syncytia. As two cells fuse and the RI at the boundary becomes similar, how well does the algorithm predict identity of two cells vs one. A supplementary movie illustrating the segmentations of cells forming syncytia will be informative.
- Improvements to figures:
 - In figure 2C and 2D, it will be easier to see the segmentation and underlying data if segmentations are shown as contours rather than masks. Currently, the opaque binary mask hides the underlying data.
 - Figure descriptions of Extended figures 3, 4, 6, 7, 8 need to be more complete. What is the difference between the three-figure sets in each panel?
- We have recently preprinted work in this area and some of our findings concur with data reported in this manuscript: <https://www.biorxiv.org/content/10.1101/2023.12.19.572435v1>. We will let the authors decide if they should cite this preprint.
- Reproducibility of code:
 - The code is not reproducible in its current state. We suggest hosting some image data, and scripts and jupyter notebooks that allow others to run inference using the pretrained models.

We recommend publication after the authors address the above input.

Co-reviewed:
Soorya Pradeep & Shalin Mehta
March 1, 2024

Reviewer #5:

Remarks to the Author:

Dynamic label-free analysis of SARS-CoV-2 infection reveals virus-induced subcellular remodeling.

Nell Saunders¹, Blandine Monel¹, Nadège Cayet², Lorenzo Archetti^{3,†}, Hugo Moreno^{3,†}, Alexandre Jeanne^{3,†}, Agathe Marguier³, Julian Buchrieser¹, Timothy Wai⁴, Olivier Schwartz^{1,5*}, Mathieu Fréchin^{3,6,*}.

Point by point answers to reviewers

REVIEWER COMMENTS

Reviewer #1 (Remarks to the Author):

I have been asked to concentrate in my review on the Bayesian network (BN) modeling aspects of this study, as it pertains to my area of expertise. Therefore, I will not comment on the overall significance/impact in the broader SARS-CoV-2 infection context. That being said, I agree with the authors that using BNs for dissecting cell organelle regulation patterns is a novel and potentially fruitful idea.

We thank reviewer #1 for her/his overall positive comments. We implemented the requested improvements, which increased the quality of our work and made our BN network usage clearer and sounder.

The BN modeling application presented in the manuscript is sound overall. However, there are caveats, some major. Below I list the issues that need to be resolved before the paper is accepted for publication:

1. The authors use the term "Bayesian statistics" throughout the manuscript. This is misleading, as BNs have little (if anything) to do with Bayesian statistics. BN is a probabilistic graphical model, part of the machine learning (and not Bayesian statistics) toolkit. I suggest the authors use "Bayesian network modeling" instead of "Bayesian statistics".

We agree with Reviewer #1 and modified the text accordingly. (line 32, 71, 689)

2. In "Bayesian network structure learning" section (lines 1018+), there is a number of issues. First, there is no "causal" learning in a strict sense. The authors should tone down the language. BNs establish directional dependencies that might suggest directional causality, but there is no causality analysis (or intervention experiments with subsequent BN re-construction) in the study.

We changed the term "causal" by "directional dependencies". (line 1226)

Second, K2 is not a scoring function. It is a search-and-score algorithm. Third, what was the actual scoring function used? The default scoring function in bnlearn is BIC, but the authors need to specify that explicitly.

We thank Reviewer #1 for spotting the mistake, the default scoring function is indeed BIC and it is now explicitly mentioned in the Methods section. (line 1234-1237)

The best BN for each condition over each dataset was defined using a greedy score-based structure learning. For this study we used the K2 and hill climb search-and-score algorithm, which incrementally tests BN alternatives to improve the default scoring function BIC that itself determines the probability of the BN structure given its training data.

3. Line 396. It's conditional dependencies, not relationships.

This is now changed and reads line 473:

but not, to the best of our knowledge, to model the hierarchy and regulation existing between cell organelles. Established BN methods⁵⁸ allow to search for the conditional dependencies and probabilities between factors of interest.

4. Why use bootstrapping? This can be very misleading with BNs because of the multicollinearity (that BNs are designed to resolve in the first place). "Averaging" the resolutions of, say, a particular V-structure in a BN over 10,000 replicates strikes me as more potentially confusing than illuminating. I believe bootstrapping is, at best, unnecessary in this context, especially given the favorable dimensionality of the data. Bootstrapping will quantify robustness, but not the actual "signal strength".

We fully agree with Reviewer #1 in that our bootstrapping approach is meant to establish confidence in the robustness of our models. We want, by doing so, to diminish the impact of difficult to detect, non random, possible technical biases that are inherent to time lapse microscopy and cell biology procedures, even if, *a priori*, dimensionality is satisfying. We augmented our methods with this point. We thank reviewer #1 to make us aware of the necessity to go further in the discussion of our technical choices (line 1228-1232):

This was performed 10000 times in a typical bootstrapping approach where each of the 10000 dataset has the same size as the original dataset and is generated by random sampling with replacement. This approach allows to capture the most robust network structure through BN averaging and to diminish the impact of potential experimental and time related technical biases.

We also agree that the signal strength quantification could be improved. We now use the averaged pairwise Chi2 score to define the edge thickness of our network conditional dependencies in figure 6. It reads line 526-529:

All edges at each iteration were validated by a chi-squared, g-squared, log-likelihood, and Cressie-Read pairwise tests all displaying a p-value $< 10^{-6}$ over discretized and non-discretized data (See extended data table 1-3 for chi-squared test details).

To avoid overloading the figure 6, we now provide 3 new tables, Extended data table 1, 2 and 3 that report, for each edge of each network, the frequency, average score, maximum degree of freedom (dof) and p values provided by the chi squared pairwise tests over all iterations. The three new table legends read line 533 to 554:

Extended data table 1 | Key metrics of unperturbed cell organelle cross regulation (OCR) network. Each edge of the OCR is described by its frequency over bootstrapping iterations, its mean chi-squared score, its maximum observed degree of freedom and p-Value.

Extended data table 2 | Key metrics of SARS-CoV-2-infected cell organelle cross regulation (OCR) network. Each edge of the OCR is described by its frequency over bootstrapping iterations, its mean chi-squared score, its maximum observed degree of freedom and p-Value.

Extended data table 3 | Key metrics of Syncytin-1-expressing cell organelle cross regulation (OCR) network. Each edge of the OCR is described by its frequency over bootstrapping iterations, its mean chi-squared score, its maximum observed degree of freedom and p-Value.

5. Following up the direct dependencies established via BN with pairwise statistical tests (chi2, in this case) is a good idea, and inspires confidence. However, was it done over the discretized variables? I assume Yes. In which case, I would suggest using different statistical tests (e.g., a logistic regression or a point-biserial, for continuous/discrete mixture) without discretization.

The statistical tests were done over both discretized and non-discretized variables. , we now add to the chi2 test a g-squared, Loglikelihood, and Cressie-Read pairwise statistical tests results, all providing p-Values below 10^{-6} . It reads line 526-529:

All edges at each iteration were validated by a chi-squared, g-squared, log-likelihood, and Cressie-Read pairwise tests all displaying a p-value $< 10^{-6}$ over discretized and non-discretized data.

Methods line 1239-1246 now reads:

We performed multiple independence test to define edge strengths of our BN using each of the three models and their related discretized and non-discretized data. For each pair of our learned BN, a chi squared, g-squared, loglikelihood and Cressie-Read statistical test is performed to determine its p-Value and associated score. The mean chi squared scores over all bootstrapped iterations are then used as a quantification of the strength of the tested edges.

6. Lines 401, 451. Again, "causal" is too strong.

The sentence now reads line 476-478:

We then reported under the form of inter-organelle networks the occurrence of specific directional dependencies.

7. (MAJOR) In line 422, the authors talk about the network's "rewiring". However, all the "rewiring" we can see is a slightly different factorization of the joint probability distribution. This could happen for any number of reasons, such as the glaringly obvious one of introducing a new node/variable (Syncytium) into the graph. Therefore, the "rewiring" conclusions (lines 425, 458) are not strongly supported. I am not saying that BNs are not useful at suggesting changes --- they are, but I'd rather use the language of "suggesting a direct link" than "identifying a direct link".

We corrected the text as suggested (lines 578) :

Our bootstrapped BN analysis (Fig. 6B) suggested a direct link between formation of the infection-induced syncytium and LDs.

We also augmented the discussion section to nuance the rewiring conclusion, it now reads at line 643-644:

Thus, the use of causal inference provides a partial view of OCR networks and of their potential rewiring upon perturbations.

8. (MAJOR) How did the authors discretize the variables? BNs are notoriously sensitive to discretization. A summary table (variables, their descriptive statistics and distributions/shapes, discretization bins) should be present in Supplementals.

Our approach relies on the Bnlearn implementation of Bayesian discretization method with quadratic complexity described in Yi-Chun Chen et al. 2015 - Learning Discrete Bayesian Networks from Continuous Data. We provide now also the result

of such discretization on the BN dataset, as an extended data figure 10 displaying the discretization and distribution of our data, for each condition and organelles.

9. The authors are talking about DAGs (directed acyclic graphs), but the graphs in Figure 6 look more like equivalence classes (with many bidirectional edges). This is fine, but should be mentioned in Figure 6 legend to avoid the readers' confusion.

This is now corrected (line 523-524). It now reads:

Averaged Bayesian networks representing the dependencies (equivalence classes) between mitochondria, nuclei, nucleoli, lipid droplets and cytosol dry mass states and the syncytial state

10. How are the edge strengths in Figure 6 determined, via bootstrap? This should be in Figure 6 legend. bnlearn has many options for edge strength, some more appropriate here than the others. I'd rather the authors use scoring criterion-based edge strengths than bootstrapping (see Comment 4 above).

We improved that point as described in comment #4.

The Figure 6 legend now reads line 526-529:

All edges at each iteration were validated by a chi-squared, g-squared, log-likelihood, and Cressie-Read pairwise tests all displaying a p-value < 10⁻⁶ over discretised and non-discretised data (See extended data table 1-3 for chi-squared test details).

In general, BNs are useful for getting a holistic view of the domain, and for hypothesis generation, but not so much for comparative hypothesis testing, as the authors are doing here with the "rewiring". I don't think that the "rewiring" analysis is convincing in itself (although the subsequent experiments by the authors do inspire some measure of confidence).

To illustrate, when two edges are replaced by one different edge (such as in Figure 6 A->B transition), the correct "apples-to-apples" analysis would be (i) to skip the bootstrap, (ii) do the pairwise statistical tests between *all* pairs (including Cytosol-LD in Figure 6A and Nuclei-LD and Mito-LD in Figure 6B), not just along the direct dependencies suggested by the BNs, and (iii) compare the p-values for all (direct and transitive) pairwise relationships in Figure 6AB networks. Then, one would have a stronger argument in support of rewiring.

We thank reviewer #1 for his/her suggestion on how to strengthen our rewiring arguments. Our analysis displays only the edges that have been passing the required statistical test (Chi-squared in this case). Thus, edges differences, what we interpreted as a possible "rewiring", should satisfy Reviewer #1 request of p-Value

based validation/comparison. We made this point clearer too in the Methods section line 1239-1245:

The mean chi squared scores over all bootstrapped iterations are then used as a quantification of the strength of the tested edges. Only edges satisfying the pairwise statistical test and observed at a frequency superior to 0.15 are displayed (see extended data table 1-3).

Reviewer #1 (Remarks on code availability):

I have reviewed the authors' application of bnlearn, which was unsatisfactory. It would be difficult for the readers to replicate the authors' BN analysis. Please see above in Comments for Authors.

We have simplified and clarified the code usage.

Reviewer #2 (Remarks to the Author):

In the present article, authors highlights the subcellular restructuring triggered by SARS-CoV-2 infection introduces alteration in cell population dynamics and cross-talk between compositions which can be studied by a fresh AI-enhanced, label-free approach for real-time examination. With meticulous attention, they documented the progression and scope of these changes in live cells, by capturing and measuring cellular and organelle dynamics in real-time, along with providing extensive evidence of the virus's direct impact on lipid droplets, mitochondria, and nuclei. The manuscript can have potential interest for the readers who plan to work in a similar field. References are updated and correctly cited in the manuscript. However, I want to express some concerns which are needed to be addressed in order to improve the scientific quality of the manuscript as well as to be considered for publication.

The page and line mentioned in the comments is on the basis of main article pdf file provided.

We are grateful to Reviewer #2 for her/his time and efforts, and overall positive comments.

Major points

1. The claim "However, we detected 142 similar mitochondrial movements in both SARS-CoV-2- and Syncytin-1-induced syncytia." Situated at line 145 page 5 is not evidently depicted in the mentioned figure (fig.1). How is this "mitochondrial movements" measured here in fig.1?

We agree that the term mitochondrial movement at this stage of the manuscript is incorrect. It was not quantitatively measured, and we believe that our qualitative observations should be indicated as such.

We corrected the text accordingly, line 158:

However, mitochondrial movements in both SARS-CoV-2- and Syncytin-1-induced syncytia seemed globally similar.

2. I don't find "Fig.1 (D) Time lapse imaging data projection for 2D computer-118 vision analysis and qualitative assessment of cells." In the figure embedded in article pdf.

We modified figure 1D caption that was misleading. It now reads line 131:

Outline of our time lapse imaging analysis pipeline ensuring data processing, computer-vision, and quantitative assessment of cells.

3. The claim in line 288-290 page 14 that "LD remained well-defined even when their size increased or when they moved from the cytosol to the perinuclear region (Fig. 3C and 3D)." is not much decipherable from figure 3C-D where nuclear staining is not described.

We added a new Extended Data Figure 7 (which shifted EDF-7 to -8 and -8 to -9) bringing more details on the LD segmentations closer to the perinuclear region. Its legend now reads lines 333-334:

Extended data figure 7 | Detection of SARS-CoV-2-induced lipid droplets in the perinuclear region of infectious syncytium.

4. Can authors discuss with probable molecular mechanism how "SARS-CoV-2 could use syncytia formation to promote mitochondrial alterations " proposed in line 322 in page 14?

We are happy to see that Reviewer #2's curiosity is stimulated by this topic, as we are. We modified the text to be more explicit, it now reads line 367-370:

The observation that Syncytin-1-expressing cells also displayed a decreased mitochondrial dry mass suggests that SARS-CoV-2 could use syncytia formation to promote mitochondrial alterations more efficiently through molecular and structural mechanisms that still need to be discovered.

And in the discussion (line 658):

This will allow to further explore the whole infection process and better characterize the isolated role of syncytia during infection such as the impact of almost instantaneous small molecules redistribution and the mixing of multiple signaling or metabolic states.

5. As there are many cellular dsRNA intermediate exists, author should clearly discuss how the specificity is ensured in the treatment mentioned in , "Anti-dsRNA antibody selectively recognizes viral RNAs and not cellular RNAs."(line 351, page 16).

We agree with Reviewer #2 that this selectivity point can be made clearer. This antibody is specifically used to distinguish infected from non-infected cells. We added a reference and modified the text. It now reads (lines 401-405):

Anti-dsRNA antibody specifically recognizes double stranded RNA (dsRNA) of greater than 40 bp in length generated during the replication of positive sense genome viruses and thus selectively recognizes viral RNAs and not cellular RNAs⁵¹, validated further by the absence of dsRNA signal in cells that do not display an NSP3 signal (Fig. 5A-C and Extended data fig. 7).

The reference 51 is : <https://pubmed.ncbi.nlm.nih.gov/16641297/>

6. A correlation scatter (e.g., pearson) plot for the collocalisation claim of "dsRNA and NSP3 "in line 353,354 page page 16 would be more consolidating.

We agree that the manuscript will benefit from an additional plot confirming that dsRNA and NSP3 signals are not collocalising. We added it as Figure 5C,. The legend and text has been changed accordingly. The legend now reads lines 432-433:

(C) Scatter plot of NSP3 against dsRNA fluorescent signal and Pearson correlation coefficient of 0.34 confirm absence of signal colocalization.

And in the main text line 405-408:

NSP-3 and dsRNA, always both present in single cells showing fluorescent signal (Fig. 5A-D), rarely colocalized within cells at the subcellular level as demonstrated by the absence of correlation of both signal (Pearson coefficient of 0.34 (Fig. 5B and C).

Minor points

1. Methodological description of techniques or protocols not being reported for the first time in this paper, e.g., HTM should be a part of "Methods" but in this case has been provided in "Results" section (pdf page 2 line 84-90, 93-97). Authors should consider

reducing the detailing in result part until it is profoundly indispensable to describe the results and the settings are different in every observations or events.

This change is now implemented. The HTM description has been moved to the methods and we have reduced the detailing in the result section. The methods reads line 1032-1045:

The 520 nm laser is divided into two beams to create an interferometer setup⁸⁰. One of the beams, referred to as the object beam, interacts with the sample before being gathered by the 60× objective lens. Simultaneously, the other beam, left unperturbed, functions as the reference beam. When meeting on a dichroic mirror the two beams create an interference pattern a.k.a. a hologram, further captured by a CMOS camera (Extended data fig. 1A). This conventional holographic technique is augmented with the rotational scanning of the sample⁸¹ (Extended data fig. 1B) using a rotating mirror that reduces the problem of coherent diffraction noise⁸² to imperceptible levels (Fig. 1B and 1C). The series of holograms are then assembled by complex field deconvolution^{83,84}. Our HTM device incorporates dynamically adjustable mirrors that enable its optics to self-regulate throughout an acquisition and accommodate variations in the sample, such as medium evaporation or variations in meniscus curvature. Combined with an automated stage and a laser based autofocus, our HTM device has stable performances over long periods of time. Finally, the HTM device possesses an epifluorescence module allowing for simultaneous fluorescence and RI imaging (Extended data fig. 1C).

Overall the present study exhibits descriptive observational narration exploiting number of online analysis tools which proposes utilizable avenues of machine learning and statistical models in finding intracellular organelle cross talk and response against a stress. They made good use of Bayesian networks (BN) to model the hierarchy and regulation existing between cell organelles which occurred to me quite novel and except of some mention regarding some nuclear bodies in the following paper (A Bayesian network model of proteins' association with promyelocytic leukemia (PML) nuclear bodies By Mikael Bodén et al. PMID: 20426694) no other mention of such kind is easily available.

We thank Reviewer #2 for her/his supportive comments. We have added this interesting reference, which is now the reference 56 and it reads line 427:

BN are an established tool for modelling biological datasets⁵¹ in fields such as signaling⁵², genomics^{53,54}, or immunology⁵⁵, and to model protein-nuclear bodies interactions⁵⁶, but not, to the best of our knowledge, to model the hierarchy and regulation existing between cell organelles.

Reviewer #3 (Remarks to the Author):

Review of Dynamic label-free analysis of SARS-CoV-2 infection reveals virus-induced subcellular remodeling.

The manuscript submitted by Saunders and co-authors that proposes a methodology to analyse images of SARS-CoV-2. The manuscript has interesting contributions in the acquisition, with label-free holo-tomographic microscopy, a deep-learning based segmentation of the images, and a series of observations derived from the segmented datasets. My expertise in the acquisition and biology is limited, thus I will concentrate on the processing of the data. Still, I am well impressed by the acquisition of the images, the quality is very good and provide excellent resolution for the subsequent segmentation and analysis. In terms of the comparisons requested, these are well documented with a large number of images to compare with.

There is a criticism of lack of novelty in the ML. I would not consider that training a well-known UNet for a specific application would count as novelty. Indeed, there have been many newer architectures based on UNet that improve times or results on certain datasets. Contribution in the quality of detection would require a numerical comparison with other methodologies. However, I would not think that it is necessary to have a contribution with a new architecture for this paper that has already provided novelty in other areas, namely the acquisition and the findings from the data itself. Thus, if UNet provides a sufficiently good segmentation, that is more than enough in this paper.

We thank Reviewer #3 for her/his overall supportive comments.

I would just highlight, that the section "Deep Learning object detection" (lines 947 onwards) should provide a better description of the architecture, number of layers, number of images with which it was trained were there patches or whole images, time to train, software used (somewhere else Python is mentioned but not in the main document).

We agree with Reviewer #3 and have added all the required information in the deep learning object detection section.

Lines 1125-1143:

The U-Net consists of an encoder-decoder structure meeting at a bottleneck layer. The encoder component has four blocks, each comprising three 3x3 convolutional layer followed by Rectified Linear Unit (ReLU) activation and batch normalization. Between these blocks, 2x2 max-pooling layers are used to down-sample the feature maps and dropout layers of 0.5 are used to prevent overfitting. The input layer is the full 448x448 image. Because of the 3x3 convolutional layer, and 2x2 max pooling layer, the following blocks are then 224x224 pixels and 32 feature maps, 112x112 and 64 feature maps, 56x56 and 128 feature maps, 28x28 and 256 feature maps.

The decoder component is symmetric to the encoder, it consists of four blocks each like the encoder blocks. However, instead of max-pooling layers, transposed convolutional layers (also known as deconvolutional layers) are utilized to up sample the feature maps. The first decoder block is in common with the last encoder block. Skip connections are established by concatenating the feature maps from the corresponding encoder part to its symmetrical decoder part. Finally, the size of the bottleneck layer is 28x28 and 512 feature maps.

Predictions in images larger than 448x448 involve doing patches of prediction of size 448x448. Such window is then moved across the image stepwise 50 by 50 pixels, horizontally and vertically and the overlapping predictions are averaged.

Reviewer #3 (Remarks on code availability):

The github page would benefit from a longer description, e.g., examples of how to run the code, figures, etc.

We have improved our GitHub page to render it more user friendly and explicit.

Reviewer #4 (Remarks to the Author):

Sauders. *et. al* report dynamic measurements of organelle remodeling in U2OS-ACE2 cells during SARS- CoV-2 infection using label-free holo-tomography (HTM). The changes in organelle morphology and composition are measured in terms of the dry mass. They integrate multiple machine learning and deep learning algorithms (random forest classifier, convolutional neural networks, and Bayesian modeling) to identify how key organelles (mitochondria, lipid droplets, nuclei, nucleoli) are remodeled during infection and suggest some causal links between phenotypes. Segmentation of organelles is guided by fluorescent markers. Some of the key findings, such as increase in the size and density of lipid droplets, are validated with complimentary electron microscopy and confocal microscopy.

Overall, above contributions report exciting advance at the interface of live cell imaging, machine learning, and infection biology. The paper also clarifies some of the strengths and weaknesses of the approach, e.g., organelles that have poor contrast in refractive index cannot be profiled with HTM, the segmentation models they report are only for 2D images, and Bayesian model they employ cannot capture feedback loops. Writing is clear and well structured.

We thank Reviewer #4 for her/his supportive comments and for pointing out their preprint. We addressed all the comments below.

We found following areas of improvement in clarity and accuracy:

- HTM is a 3D imaging system. Authors should report z flythroughs from the apical to basal side of the cells in the infected and uninfected conditions and compare them with 2D maximum projection. What is the effect of 2D maximum projection on the estimate of the dry mass? We'd argue that data in Fig. 4B and 4D cannot be interpreted in the units of pg or pg/um², since the measurements are done from the maximum projection rather than actual reconstruction. For example, what is the percentage error in dry mass measurement of nuclei due to endoplasmic reticulum on top of it which is not distinguished.

We thank Reviewer #4 for his comment. We fully this point of view. While absolute certainty is out of reach, comparisons between similarly established control and perturbations experiments is the best practice in microscopic image quantification. We make the methods clearer on this point, it now reads line 1116-1119:

Custom routines were used to convert 3D volumes into 2D maximal projections along the z-axis from -2 to +6 μm around the point of focus, which ensures to capture entire cells and their organelles while those biological objects may move along the z axis.

We also correct an error in our units, in fact, we should report dry mass densities as $\text{pg}/\mu\text{m}^3$ since a single slice has a volume of $X \times Y \times \text{typical voxel volume}$ ($0.2 \times 0.2 \times 0.4 \mu\text{m}^3$) and not simply a surface. We apologize for this error and have corrected it in figure 4.

A full, direct comparison of dry mass in 3D volumes compared to a projection would require the development of dedicated 3D deep learning models and is a major challenge for a future study of similar scale, moreover, 3D reconstructions readily carry uncertainties and so will the derived quantifications (see below), requiring anyways for the same controlled approach that we have used here.

We nonetheless improved our manuscript with an extended data figure 11 that establishes at a reasonable scale that our dry mass density calculations based on projections recapitulate well a full 3D derived dry mass density calculation from a manual 3D cell segmentation. The new figure legend reads line 680-686:

Extended data figure 11 | Maximum intensity projection images recapitulate of holotomographic 3D volumes recapitulate volume-derived dry mass densities. (A) Holotomographic microscopy (HTM) returns 3D refractive index maps (B) are transformed into maximal intensity projection images. (C) The dry mass densities calculated from manual 3D cell segmentations, or automated 2D cell segmentations provide similar results, confirming that 2D maximal intensity projections recapitulate well the whole cell content.

We also added a discussion on the limitations of our work at this level, on the necessity to use 2D projections for feasibility and robustness, and the importance to place our measures in the appropriate context. It now reads line 660-671:

Fourthly, to ensure capturing all biological structures despite dynamic variations in z distribution, we work with a typical 2D projection procedure that is applied similarly on all 3D sample reconstructions. Our quantifications therefore are approaching the true dry mass content rather than being absolute (Extended data fig. 11). It is important also to mention that a full 3D-based quantification is not an absolute reference either, even with the best possible microscopic device, absolute quantitative knowledge is out of reach. This is why controlled experiments similar to those made in our study, are and will remain essential. Importantly, a 3D-based quantitative investigation at similar experimental scale represents a massive technical and computational challenge that is the matter of future studies.

Please note that ER is not visible in our images. See Sandoz et al. 2019 PLOS Biol.

- How the cytosol was segmented requires an elaboration: The authors explain that a sharpening of the outlines was performed by object propagation within the RI signal for cytosol segmentation.
- Validation of cytosol segmentation same as other segmented organelles is not presented.

We thank Reviewer #4 for pointing out the lack of clarity at this level. In fact, all cell segmentations are shown and can be checked in our supplementary movies under the form of red outlines. The cytosol was defined by the subtraction from the cell segmentation mask of all other organelles segmentations. In other terms, the cytosol is defined here as everything in the cell that is not mitochondria, lipid droplet, nucleus or nucleoli. We improved the methods

section to make that clearer, as well as the body of the text mentioning the cytosol segmentation.

It now reads in the methods: lines 1210-1212:

Definition of the cytosol compartment:

The cytosol compartment is defined as the mask created by the subtraction of all organelles masks (Nucleus, nucleoli, lipid droplets, mitochondria) from the cell segmentation mask.

And in the main text it reads line 291-292:

The cytosol was defined by the removal of all organelle segmentations from the single cells segmentations. (Extended data figure 6 and extended data videos 1 to 19).

What is the ground truth in this case? Is it generated manually by experts?

We apologize if the information was not clear to Reviewer #4, the direct answer to this point is already present in the manuscript at line 1156-1157:

The ground truths for the cells, nuclei, and nucleoli, were made semi-manually with the help of a custom labeling tool and the guidance of fluorescently stained cells.

This section does not mention the cytosol as it is strictly speaking not a mask that we generated from a direct model's prediction but as mentioned above, a product of masks subtractions, we therefore added this precision at line 1152-1153:

No ground truths were generated for the cytosol compartment as this mask type is generated from subtracting organelles masks from the cell mask, as described below in dedicated section.

○ How well does this algorithm work as authors track formation of syncytia. As two cells fuse and the RI at the boundary becomes similar, how well does the algorithm predict identity of two cells vs one. A supplementary movie illustrating the segmentations of cells forming syncytia will be informative.

We thank Reviewer #4 for pointing out the lack of clarity at this level. All cell segmentations are shown in our extended data movies and extended data figure 6 aims at showing single cells / syncytium segmentation quality and we thus feel those material specifically bring the information Reviewer #4 requested. We made the text clearer so the reader know that such transition can be verified there.

It now reads (288-290):

We used our algorithms to identify over time single or fused cells with a precise segmentation even in time of transition from the single cell to syncytium state (Extended data fig. 6 and extended data videos 1-19)

• Improvements to figures:

○ In figure 2C and 2D, it will be easier to see the segmentation and underlying data if segmentations are shown as contours rather than masks. Currently, the opaque binary mask hides the underlying data.

The underlying data is visible in zoomed-in panels and in the fluo channel image where the fluorescence signal color is made transparent. We changed the figure legend to make that point clearer, it now reads line 193-195:

(C-F) Comparison of (C) nuclei, (D) nucleoli, (E) lipid droplet and (F) mitochondria detection within refractive index images with their respective fluorescent label signal made partially transparent such that underlying cellular structure are visible.

- Figure descriptions of Extended figures 3, 4, 6, 7, 8 need to be more complete. What is the difference between the three-figure sets in each panel?

We have improved the respective figure captions. They now read:

Extended data figure 3 | Comparison of organelle fluorescent signal and machine learning-based organelle prediction. (A) Various forms of Nuclei, (B) nucleoli, (C) lipid droplets and (D) mitochondria, are detected efficiently in cells of various forms and confluency. For metrics quantifying the quality of our predictions see figure 2G.

Extended data figure 4 | Machine learning detects unperturbed or fragmented mitochondria after silencing of *OPA1*. Control U2OS cells display long, tubular, connected mitochondria, while U2OS cells experiencing *OPA1* silencing have punctuated, over-fragmented mitochondria. Zooms show that mitochondria are detected across a wide range of various morphologies.

Extended data figure 6 | Deep learning-based label-free cell segmentation detects single cells and syncytium with precision. At 0 and 9 hours, we can see that cells are all efficiently detected. At 27 and 36 hours, one can observe that the large syncytium is well segmented as well as the co-existing single cells that did not fuse with the syncytium, validating our proper cell vs syncytium segmentation robustness. All our cell vs syncytium proper segmentations can be verified in our extended data videos 1-19.

Previously Extended data figure 7:

Extended data figure 8 | Perinuclear lipid droplet accumulation is a marker of infection. Each of the five panels show that cells with large lipid droplets all show NSP3 and dsRNA immuno-fluorescent signal, while those with no lipid droplets do not display NSP3 and dsRNA immuno-fluorescent signal. See also figure 5.

Previously Extended data figure 8:

Extended data figure 9 | Lipid droplets of infected cells are not surrounded by mitochondria. The four U2OS cells (cell 1, cell 2, cell 3, cell 4) and syncytium infected with the SARS-CoV-2 Wuhan strain and developing large lipid droplets do not show any mitochondria close to lipid droplets when observed by electron microscopy. See also figure 5E and 5F.

- We have recently preprinted work in this area and some of our findings concur with data reported

in this manuscript: <https://www.biorxiv.org/content/10.1101/2023.12.19.572435v1>. We will let the

authors decide if they should cite this preprint.

We thank Reviewer #4 for sharing this preprint. We apologize for the omission which was not intentional. It is very relevant explaining how label-free microscopy will help quantitative biology. We are happy to cite it as citation 68, lines 618-619 it reads:

Among currently available label-free techniques, holotomographic microscopy (HTM) produces images whose quality enable effective computer vision solutions^{20,68}.

Reviewer #4 (Remarks on code availability):

The code is not reproducible in its current state. We suggest hosting some image data, and scripts and jupyter notebooks that allow others to run inference using the pretrained models.

We improved our code accordingly.

Reviewer #5 (Remarks to the Author):

We thank Reviewer #5 for her/his time and efforts in reviewing our manuscript together with a co-Reviewer.

Reviewers' Comments:

Reviewer #1:

Remarks to the Author:

The revised manuscript is significantly improved (with regard to BN analysis). However, I suggest two modifications before this work is ready for publication:

1. Page 21, line 408: replace "conditional dependencies" with "direct dependencies (while filtering out spurious correlations)"

2. Analysis in the Extended Data Tables 1-3 is not what I was asking for in the previous round of reviews. The fact that the recovered edges in the BNs are highly supported by pairwise statistical tests is hardly surprising and is to be expected. It is nice to have, but is not really needed.

What is needed here is different --- the authors' argument hinges on the BN's rewiring between Fig. 6B and Fig 6C. Specifically, Cy->Sy relationship is mediated by LD in Fig 6B, but is direct in Fig 6C, with LD node becoming "orphaned". The BNs in Fig. 6 are probably sufficiently robust to make such a conclusion, but additional pairwise statistical tests would be helpful in supporting this claim. Therefore, instead of Extended Data Tables 1-3 (which could go into supplementals), the authors should do pairwise tests for **all** pairwise combinations (Sy-LD, Cy-Sy, Cy-LD) under **both** scenarios (Fig 6B and C), and demonstrate that Sy-LD and Cy-LD are (presumably) stronger in B than in C, and Cy-Sy is (presumably) weaker in B than in C. Now, B and C datasets are not directly commensurate, so the above analysis is not exactly dispositive, but it will be (presumably) a convincing piece of additional evidence in support of the authors' conclusion.

Reviewer #2:

Remarks to the Author:

My comments are well addressed and amendments are duly incorporated in the manuscripts and/or figures. The responses to my comments are satisfactory.

Reviewer #3:

Remarks to the Author:

All the issues that I raised have been properly addressed.

Reviewer #4:

Remarks to the Author:

We thank the authors for a thorough revision of the manuscript. All of our major concerns are addressed and we don't think further revision of figures and text is needed. We recommend publication.

Reviewer #5:

Remarks to the Author:

Point by point answers to reviewers

Extended data table 1 | Pairwise statistical tests strengthen key organelle cross regulation (OCR) rewiring differences between infected cells and cells expressing Syncytin 1. Chi-squared score and p-values reinforce the observation that cytosol and syncytium edges towards LD exists when cells are infected by SARS-CoV-2 but not when cells only express Syncytin 1.